# Structural basis of phosphatidylcholine recognition by the C2–domain of cytosolic phospholipase A$_2\alpha$

Yoshinori Hirano[1,2†‡], Yong-Guang Gao[3†], Daniel J Stephenson[4,5], Ngoc T Vu[4], Lucy Malinina[3], Dhirendra K Simanshu[1§], Charles E Chalfant[5,6,7*], Dinshaw J Patel[1*], Rhoderick E Brown[3*]

[1]Structural Biology Program, Memorial Sloan-Kettering Cancer Center, New York, United States; [2]Graduate School of Biological Sciences, Nara Institute of Science and Technology (NAIST), Takayama, Japan; [3]Hormel Institute, University of Minnesota, Austin, United States; [4]Department of Biochemistry and Molecular Biology, Virginia Commonwealth University Medical Center, Richmond, United States; [5]Department of Cell Biology, Microbiology and Molecular Biology, University of South Florida, Tampa, United States; [6]Research Service, James A. Haley Veterans Hospital, Tampa, United States; [7]The Moffitt Cancer Center, Tampa, United States

**\*For correspondence:**
cechalfant@usf.edu (CEC);
pateld@mskcc.org (DJP);
reb@umn.edu (REB)

[†]These authors contributed equally to this work

**Present address:** [‡]Laboratory of Protein Structural Biology, Graduate of Pharmaceutical Sciences, The University of Tokyo, Tokyo, Japan; [§]Frederick National Laboratory for Cancer Research, Frederick, United States

**Competing interests:** The authors declare that no competing interests exist.

**Abstract** Ca$^{2+}$-stimulated translocation of cytosolic phospholipase A$_2\alpha$ (cPLA$_2\alpha$) to the Golgi induces arachidonic acid production, the rate-limiting step in pro-inflammatory eicosanoid synthesis. Structural insights into the cPLA$_2\alpha$ preference for phosphatidylcholine (PC)-enriched membranes have remained elusive. Here, we report the structure of the cPLA$_2\alpha$ C2-domain (at 2.2 Å resolution), which contains bound 1,2-dihexanoyl-*sn*-glycero-3-phosphocholine (DHPC) and Ca$^{2+}$ ions. Two Ca$^{2+}$ are complexed at previously reported locations in the lipid-free C2-domain. One of these Ca$^{2+}$ions, along with a third Ca$^{2+}$, bridges the C2-domain to the DHPC phosphate group, which also interacts with Asn65. Tyr96 plays a key role in lipid headgroup recognition via cation–π interaction with the PC trimethylammonium group. Mutagenesis analyses confirm that Tyr96 and Asn65 function in PC binding selectivity by the C2-domain and in the regulation of cPLA$_2\alpha$ activity. The DHPC-binding mode of the cPLA$_2\alpha$ C2-domain, which differs from phosphatidylserine or phosphatidylinositol 4,5-bisphosphate binding by other C2-domains, expands and deepens knowledge of the lipid-binding mechanisms mediated by C2-domains.

## Introduction

Lipids play indispensable roles in signal transduction, while also serving as essential structural components of the cell membrane, as energy resources, and as metabolites for the generation of hormones and eicosanoids. Phospholipase A$_2$ (PLA$_2$) is a member of a diverse enzyme superfamily that hydrolyzes the *sn-2* acyl bond of glycerol-based phospholipids (*Smith, 1989*; *Dennis et al., 2011*). Cytosolic PLA$_2\alpha$ (cPLA$_2\alpha$), a Group IV mammalian PLA$_2$ family member, preferentially releases arachidonic acid from PLs in a cytosolic Ca$^{2+}$-concentration-dependent manner (*Clark et al., 1991*; *Shimizu et al., 2006*; *Leslie et al., 2010*; *Vasquez et al., 2018*). Arachidonic acid generated by cPLA$_2\alpha$ is a precursor of pro-inflammatory eicosanoids, including certain prostaglandins and leukotrienes. Consequently, cPLA$_2$-mediated bioactive lipid production plays a major regulatory role in physiological and pathogenic processes (*Bonventre et al., 1997*; *Uozumi et al., 1997*; *Leslie, 2015*).

Insights into cPLA$_2\alpha$ activation by regulatory mediators are of great importance because arachidonic acid release by cPLA$_2\alpha$ at the membrane surface is the rate-limiting step in eicosanoid

production. The ensuing prostaglandin and leukotriene production occurs via cyclooxygenases and lipoxygenases, respectively. Increases in intracellular $Ca^{2+}$ concentration that are induced by extracellular stimuli activate $cPLA_2\alpha$ by inducing translocation from the cytoplasm to the perinuclear region, (i.e. the Golgi apparatus, nuclear envelope and endoplasmic reticulum) (*Evans et al., 2001*). Mechanistically, the membrane translocation of $cPLA_2\alpha$ is driven primarily by its N–terminal C2-domain rather than its catalytic domain (*Nalefski et al., 1994*; *Davletov et al., 1998*; *Dessen et al., 1999*). Complexation of two $Ca^{2+}$ ions by the C2-domain neutralizes several Asp residues, facilitating protein docking and penetration into the membrane interface region (*Davletov et al., 1998*; *Dessen et al., 1999*; *Perisic et al., 1998*; *Bittova et al., 1999*). Occupation of one $Ca^{2+}$-binding site exerts stronger effects than occupation of the other in terms of stabilizing the membrane partitioning of the $cPLA_2\alpha$ C2-domain (*Bittova et al., 1999*; *Stahelin and Cho, 2001a*).

In addition to $Ca^{2+}$, $cPLA_2\alpha$ activators include specific lipids. Mutational functional analyses have revealed that ceramide-1-phosphate (C1P), a bioactive sphingolipid generated by ceramide kinase in the *trans*-Golgi, enhances enzyme translocation to perinuclear regions (*Pettus et al., 2004*; *Stahelin et al., 2007*; *Lamour et al., 2009*) by binding directly to the C2-domain. Phosphatidylinositol 4,5-bisphosphate ($PI(4,5)P_2$) also activates $cPLA_2\alpha$, but independently of intracellular $Ca^{2+}$ concentration (*Mosior et al., 1998*; *Das and Cho, 2002*; *Casas et al., 2006*), by binding a site that is enriched in cationic residues in the catalytic domain (*Das and Cho, 2002*; *Six and Dennis, 2003*; *Tucker et al., 2009*). The cationic residue clusters that the $cPLA_2\alpha$ C2-domain uses to bind C1P for membrane targeting (*Stahelin et al., 2007*; *Ward et al., 2013*) differ from those used by other C2-domains [e.g., protein kinase C (PKC), Syt and Rabphilin] that bind $PI(4,5)P_2$ (*Honigmann et al., 2013*; *Guillén et al., 2013*).

Structural insights into $cPLA_2\alpha$ interaction with lipids are limited. NMR data have enabled the identification of several residues in the C2-domain calcium binding loops (CBLs) that interact with dodecylphosphocholine micelles (*Xu et al., 1998*). Hydrogen-deuterium exchange mass spectrometry and molecular dynamic studies have also helped to map the membrane interaction regions of the C2-domain and have shed light on catalytic domain conformational accommodation of methyl arachidonoyl fluorophosphonate, a $cPLA_2\alpha$ active-site inhibitor and 1-palmitoyl-2-arachidonoyl-*sn*-glycero-3-phosphocholine (PAPC) substrate (*Burke et al., 2008*; *Cao et al., 2013*; *Mouchlis et al., 2015*). Nonetheless, the lack of crystal structures for the $cPLA_2\alpha$ C2-domain or for the $cPLA_2\alpha$ catalytic domain containing bound phosphoglyceride has hampered understanding of the structural basis that underlies the lipid activation mechanism(s) and the known preference of $cPLA_2$ for phosphatidylcholine (PC)-enriched membranes.

Here, we report the X-ray crystal structure of the $cPLA_2\alpha$ C2-domain bound to 1,2-dihexanoyl-*sn*-glycero-3-phosphocholine (DHPC). In contrast to the two bound $Ca^{2+}$ ions reported in the lipid-free structure (*Dessen et al., 1999*; *Perisic et al., 1998*), we observed three $Ca^{2+}$ ions coordinated in the PC-bound structural complex, consistent with the established $Ca^{2+}$ dependence of membrane interaction by the C2-domain. Tyr96 is found to play a major role in the lipid recognition and selectivity of DHPC via cation–π interaction with the lipid's trimethylammonium $[N^+(CH_3)_3]$ group. Two of the three bound $Ca^{2+}$ ions provide bridging interactions between the C2-domain and the DHPC phosphate group, which also interacts directly with Asn65. The DHPC-binding mode of the $cPLA_2\alpha$ C2-domain differs substantially from that of the $PKC\alpha$ C2-domain or the Syt1 C2A-domain bound to phosphatidylserine (PS) or $PI(4,5)P_2$, thereby expanding and deepening our knowledge of the lipid-binding mechanisms that are mediated by the C2-domain.

## Results

### Overall structure of the $cPLA_2$ C2-domain bound to DHPC

To elucidate the mechanism of phosphoglyceride recognition by the C2-domain of $cPLA_2\alpha$, we initially attempted to generate complexes of the human recombinant protein with various lipids including PC, related phosphoglycerides and C1P analogs. Despite extensive crystallization trials, the resulting C2-domain crystals contained two bound $Ca^{2+}$ ions, but no bound lipid [as also reported by *Perisic et al., 1998*]. However, using purified *Gallus gallus* (chicken) $cPLA_2\alpha$ C2-domain (81% identical and 93% highly conserved sequence relative to human) (*Figure 1A* and *Figure 1—figure supplement 1*), we obtained crystal complexes with 1,2-dihexanoyl-*sn*-glycero-3-phosphocholine

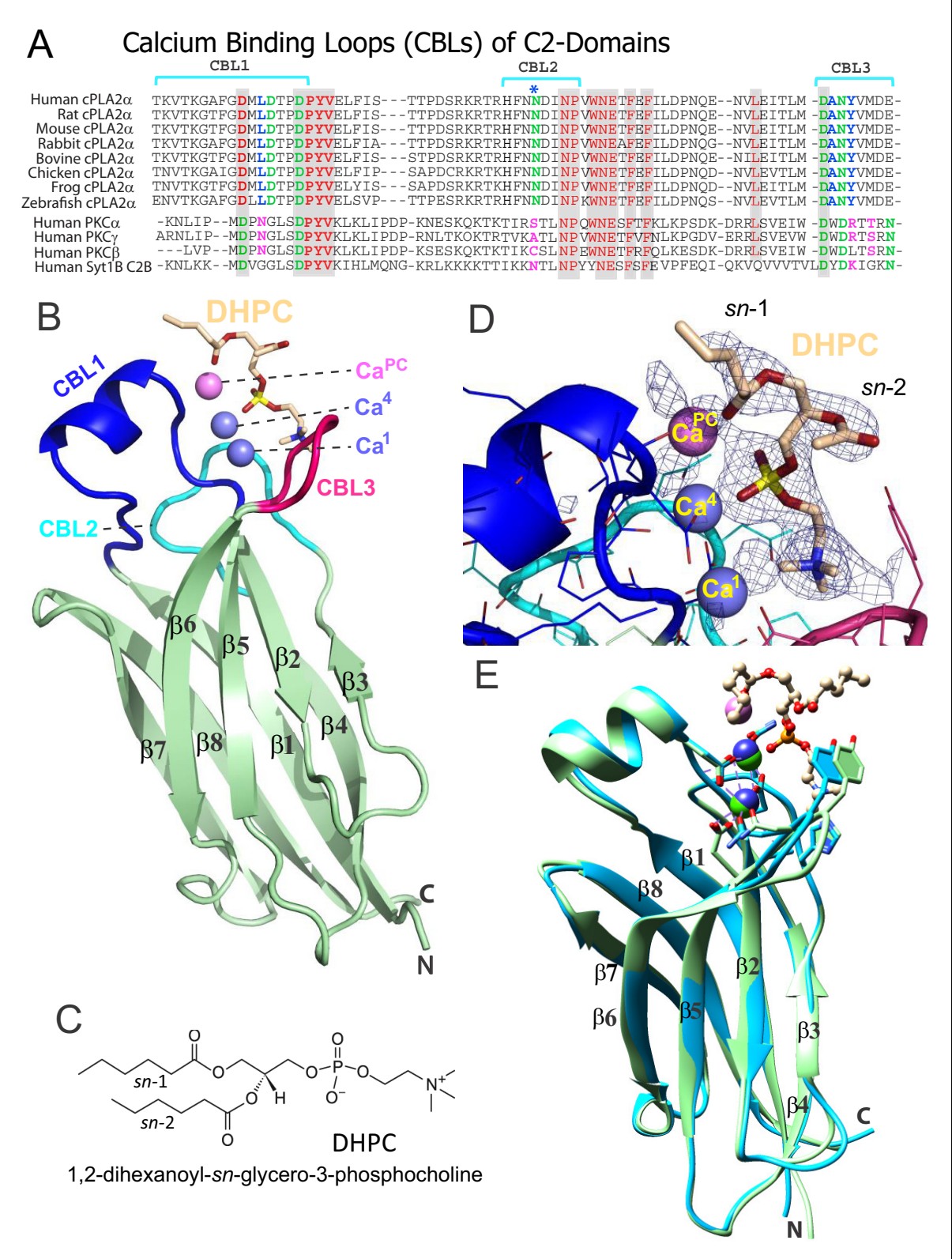

**Figure 1.** Structure of cPLA2α C2–domain containing bound DHPC and calcium. (A) Sequence alignment of C2-domain calcium-binding loop (CBL) regions in cPLA2α from different eukaryotes compared to human PKCs and Syt1. Residues that bind Ca²⁺ are green. Residues interacting directly with PC in our structural complex (blue or blue asterisk) are absolutely conserved among eukaryotic cPLA2α proteins but not in PKCs and Syt1. Conversely, residues that interact with PS in the PKCα-PS structure (magenta) are highly conserved in PKCs and Syt1, but not in cPLA2α. Shaded residues are

*Figure 1 continued on next page*

*Figure 1 continued*

identical. The human and chicken cPLA₂α CBL sequences are 92% identical and 94.5% highly conserved (see *Figure 1—figure supplement 1* for full-length sequence alignment). (B) Ribbon structure representation of the cPLA₂α C2-domain bound to 1,2-dihexanoyl-*sn*-glycero-3-phosphocholine (DHPC). The DHPC molecule (beige stick) straddles the β1–β2 loop (CBL1, blue), β3–β4 loop (CBL2, cyan) and β5–β6 loop (CBL3, red). Ca1 and Ca4 (blue spheres) are in a similar position in the apo-form structure; whereas Ca$^{PC}$ (magenta sphere) is unique to the DHPC-bound form. (C) DHPC structural formula. (D) *Fo-Fc* omit electron density map for the bound DHPC molecule at the 2.5σ contour level. (E) Superimposition of the chicken cPLA₂α C2-domain with bound DHPC (colored as in *Figure 1B*) on the human lipid-free structure (PDB: 1RLW, cyan). Root mean square deviation = 0.7 Å after superimposition of Cα atoms.

The online version of this article includes the following figure supplement(s) for figure 1:

**Figure supplement 1.** Sequence alignment of cPLA₂α for human, mouse, and chicken proteins.

**Figure supplement 2.** Tubular topology formed in the crystal lattice of the cPLA₂α C2-domain–DHPC structural complex.

(DHPC) (*Figure 1C*), enabling structure determination at 2.2 Å resolution (*Figure 1B*). The electron density map for the entire C2-domain polypeptide chain is visible except for the N-terminal glycine, a cloning residue artifact. Notably, extra electron density corresponding to a bound DHPC molecule is found in the map (*Figure 1D*), near the position reported for a bound MES [2-(N-morpholino) ethanesulfonic acid] buffer molecule in the structure of full-length human cPLA₂α (*Dessen et al., 1999*). Three C2-domain molecules comprise the asymmetric unit and each C2-domain contains one bound DHPC molecule. The structures of the individual complexes are essentially the same. The C2-domain–DHPC complex exhibits a β-sandwich topology formed by a pair of four-stranded antiparallel β-sheets (one formed by the β4, β1, β8 and β7 strands; the other by the β3, β2, β5 and β6 strands) (*Figure 1B*). Clearly resolved are the Ca$^{2+}$-binding loops (CBL1, CBL2, and CBL3) formed by the β1–β2, β3–β4, and β5–β6 loops, respectively. CBL1 contains a short α-helix. The structure of the chicken C2-domain with bound DHPC is almost the same as that of human lipid-freeform (*Dessen et al., 1999*; *Perisic et al., 1998*) with a small overall root mean square (r.m.s.) deviation of 0.7 Å after superimposition of Cα atoms (*Figure 1E*). Interestingly, we observed three bound Ca$^{2+}$ ions in the C2-domain–DHPC complex (*Figure 1B, D and E*), two of which correspond to the bound Ca1 and Ca4 in lipid-free human protein reported previously (*Dessen et al., 1999*; *Perisic et al., 1998*). [Note: The *Rizo and Südhof, 1998* numbering system is used for bound Ca$^{2+}$ in C2-domains (*Corbalan-Garcia and Gómez-Fernández, 2014*).] We found that the Ca1 ion is bound via side-chain interactions with Asp40, Asp43, Asp93 and Asn95 and the main chain carbonyl group of Ala94; whereas Ca4 is bound via interaction with the side chains of Asp40, Asp43 and Asn65 and the main-chain carbonyl group of Thr41 (*Figure 2A*). The Ca1 and Ca4 coordination networks are almost the same as those in the lipid-free structure (*Dessen et al., 1999*; *Perisic et al., 1998*). Binding of the third Ca$^{2+}$ involves a stabilizing contact with Asn65 and interaction with the DHPC phosphoryl group. This Ca$^{2+}$ is designated Ca$^{PC}$ because of its novel location at CBL1, which is unique when compared with Ca$^{2+}$-binding sites in various other C2-domains (*Rizo and Südhof, 1998*; *Corbalan-Garcia and Gómez-Fernández, 2014*).

## PC recognition

The DHPC polar headgroup docks with the CBLs, whereas the fatty acid chains are largely exposed to solvent. This orientation of bound PC in the complex is consistent with a function as an embedded membrane-anchoring element for the docking site of the protein. In the C2-domain–DHPC complex (*Figure 2B*), the DHPC *sn*-2 chain is largely disordered except for the ester group, but nearly all of the *sn*-1 chain is observable because of the stabilizing interaction with Ca$^{PC}$ and the partial contact by Leu39 (*Figure 1D*). The shorter fatty acid chains of DHPC compared to those of natural PCs may partially limit insights into the hydrophobic interactions mediated by fatty acid chains. Yet, it is noteworthy that the bulk of natural long-chain fatty acyl chains need not interact with the protein, but rather are expected to remain embedded in the bilayer to stabilize the PC headgroup docking sites in the membrane for cPLA₂α C2-domains.

The DHPC phosphoryl group is positioned at the center of the CBLs, with the -N$^+$(CH₃)₃ group of DHPC directed towards CBL3 (*Figure 2B*). The phosphoryl group is stabilized mainly by Ca$^{2+}$-mediated bridging interactions. The only direct interaction between the phosphoryl group and the C2-domain is a hydrogen bond with the side chain of Asn65 (~3.1 Å) (*Figure 2B*). Although Ca1 does not coordinate with the DHPC molecule, Ca4 does interact with two oxygen atoms of the DHPC

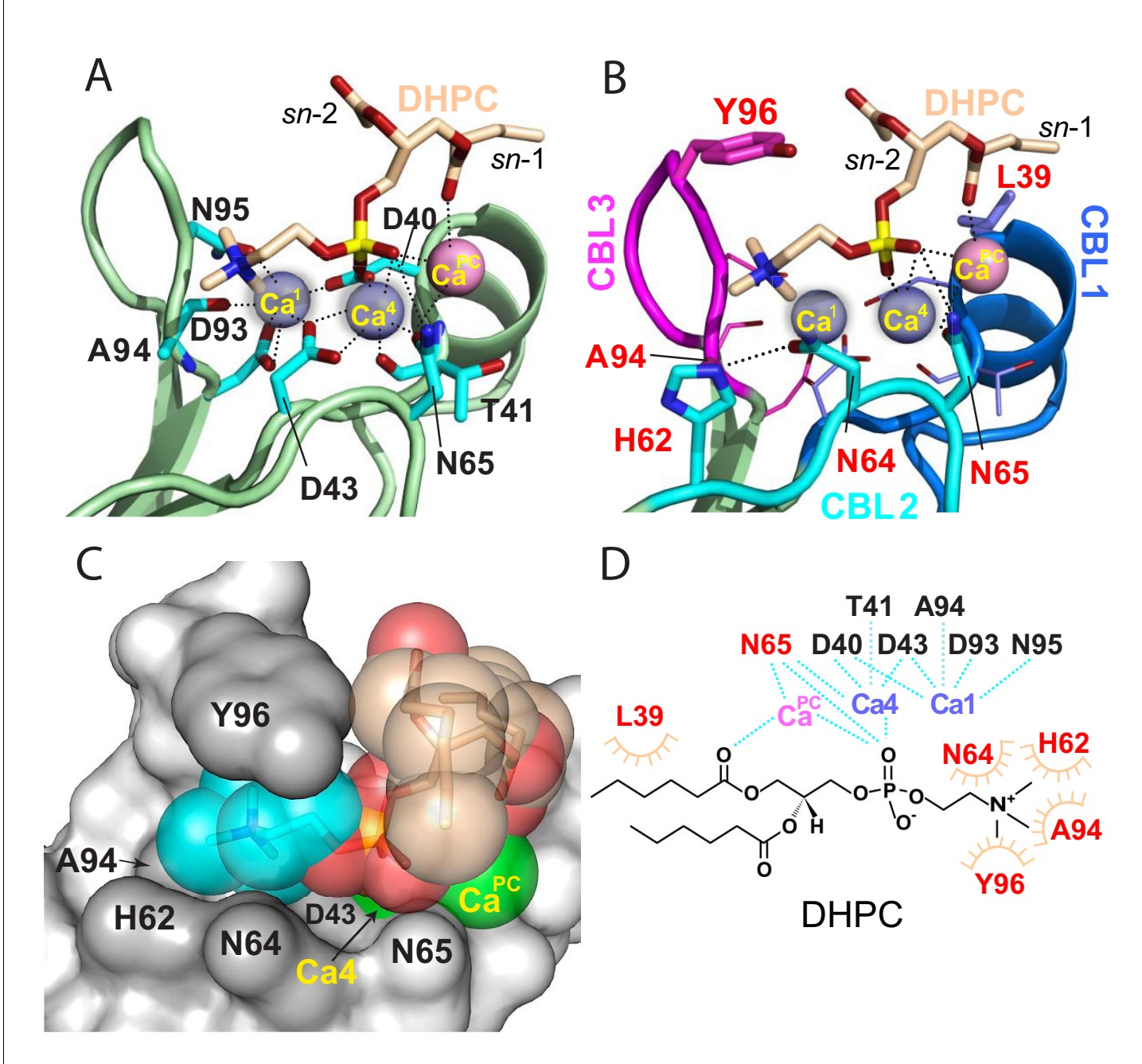

**Figure 2.** Structural interactions of the cPLA$_2\alpha$ C2-domain complexed with Ca$^{2+}$ and DHPC. (**A**) Coordination of three bound Ca$^{2+}$ ions observed in the C2-domain–DHPC complex. Residues that interact with Ca$^{2+}$ ions are labeled in black with their side-chains (cyan) depicted in a stick representation. (**B**) Same view as in panel (**A**), but with PC-mediated interactions highlighted. Residues that interact directly with DHPC are labeled in red. (**C**) Space-filling view of bound DHPC and of Ca4 and Ca$^{PC}$ in the cPLA$_2\alpha$ C2-domain. Darker gray residues (Y96, A94, H62, N64, and D43) provide contact surfaces for choline (cyan). Phosphorus is represented in orange; calcium in green; oxygen in red; and acyl carbons in beige. (**D**) Schematic summary of DHPC- and Ca$^{2+}$-binding interactions with the cPLA$_2\alpha$ C2-domain.

The online version of this article includes the following figure supplement(s) for figure 2:

**Figure supplement 1.** Zoomed views of the cPLA$_2\alpha$ C2-domain complexed with Ca$^{2+}$ and DHPC.

phosphoryl group in bifurcated fashion (2.1 and 3.7 Å). $Ca^{PC}$, which was not observed in two previous lipid-free structures (*Perisic et al., 1998*), bridges the C2-domain to DHPC via its phosphoryl group (3.6 Å) and *sn*-1 carbonyl group (3.2 Å) and also interacts with the Asn65 side-chain (3.1 Å) (*Figure 2*; *Figure 2—figure supplement 1*; *Table 1*). Thus, Ca4 and $Ca^{PC}$ mediate both partial charge neutralization of the DHPC phosphoryl group, and by doing so, bridge the C2-domain and PC. The findings further elucidate the role played by bound $Ca^{2+}$ in mediating nonspecific membrane interaction while revealing direct interaction with PC. Zoomed views of the $cPLA_2\alpha$ C2-domain complexed with $Ca^{2+}$ and DHPC are provided in *Figure 2—figure supplement 1* and associated interaction distances are summarized in *Table 1*.

The $-N^+(CH_3)_3$ group of DHPC is partially surrounded by Tyr96 of CBL3 and His62 and Asn64 of CBL2 (*Figure 2B and C*). Importantly, the Tyr96 aromatic ring stacks in planar-like fashion with the cationic $-N^+(CH_3)_3$ group. This kind of electrostatic interaction, that is cation-π interaction, occurs at distances of less than 6.0 Å between a positively charged atom or group and the flat face of an aromatic ring that has a partial negative charge due to delocalized π electrons (*Dougherty, 2013*; *Gallivan and Dougherty, 1999*). The position of the $-N^+(CH_3)_3$ group is further stabilized by van der Waals contacts with the Ala94 methyl group and by a possible weak (off-angle) cation-π interaction with the His62 imidazole ring (*Figure 2B and C*). The orientation of His62 appears to be affected by hydrogen bonding (2.8 Å) between its imidazole group and the side-chain carbonyl group of Asn64, which also interacts (weakly) with the $-N^+(CH_3)_3$ group. Notably, in the structure of the lipid-free C2-domain (*Perisic et al., 1998*), a cadmium ion from the crystallization buffer was localized between Tyr96 and His62, consistent with cation-π interaction (*Figure 1E*). Previous mutational analyses support His62 interaction with PC as well as a more significant role for this residue in binding C1P (*Ward et al., 2013*). In other proteins that specifically bind PC, such as the PC transfer protein,

**Table 1.** Interaction distances in $cPLA_2\alpha$ C2-domain.

Interaction distances (Å) associated with bound calcium in the lipid-free $cPLA_2\alpha$ C2-domain structure (2.4 Å resolution; PDB 1RLW) of *Perisic et al. (1998)* and with bound calcium and DHPC in the C2-domain–DHPC crystal complex (2.2 Å resolution) of the present study.

| 1RLW | Ca1 | Ca4 |
|---|---|---|
| Asp40 | 2.3/3.4 | 2.3 |
| Asp43 | 2.1 | 2.6/2.2 |
| Asp93 | 2.7/2.5 | |
| Asn65 | | 2.1 |
| Asn95 | 2.2 | |

| C2/DHPC | C2 | | | DHPC | | | | |
|---|---|---|---|---|---|---|---|---|
| | Ca1 | Ca4 | $Ca^{PC}$ | $N^+(CH_3)_3$ | $PO_4$ | *sn*-2 C=O | *sn*-1 C=O | *sn*-1 chain |
| Asp40 | 2.4/3.4 | 2.4 | | | | | | |
| Asp43 | 2.3 | 2.6/2.7 | | | | | | |
| Asp93 | 2.7/2.8 | | | | | | | |
| Asn65 | | 2.4 | 3.1 | | | | | |
| Asn95 | 2.3 | | | | | | | |
| Tyr96 | | | | ~4.0 | | | | |
| Ala94 | | | | 3.6 | | | | |
| His62 | | | | ~5.0 | ~8.5 | | | |
| Asn64 | | | | 3.6 | | | | |
| Leu39 | | | | | | | | 5.4 |
| Ca1 | | | | 5.5 | 5.7 | | | |
| Ca4 | | | | 6.3 | 2.1 | | | |
| $Ca^{PC}$ | | | | 8.5 | 3.6 | 3.2 | 3.1 | |

cation–π interactions involving Tyr and Trp are determinants of PC specificity (*Roderick et al., 2002*; *Kang et al., 2010*).

In our study, DHPC binding induced no major conformational changes in the C2-domain CBLs. Notably, NMR studies of the cPLA$_2\alpha$ C2-domain depict the Tyr96 indole ring in an outward position when not interacting with dodecylphosphocholine micelles (*Xu et al., 1998*). Thus, PC binding could require local conformational changes, such as inward flipping of Tyr96, to optimize π–cation interaction for complex formation.

## Functional mutagenesis analyses of PC-interacting residues in the cPLA$_2\alpha$ C2-domain

The central importance of Tyr96 for PC selectivity is supported by functional mutagenesis of the C2-domain residues observed interacting with DHPC. The Y96A point mutant, which is unable to undergo cation–π interaction with the -N$^+$(CH$_3$)$_3$ group of DHPC, exhibited significantly reduced affinity for PC bilayer vesicles when compared to either the conservatively mutated Y96F (which supports strong cation-π interaction) or the control C2–domain, as shown by SPR (*Figure 3A*; *Table 1*). Also, disruption of the PC phosphate group interaction with N65 by point mutation to Asp (N65D) significantly reduced partitioning to PC vesicles (*Figure 3A*; *Table 1*). In previous surface plasmon resonance (SPR) and DHPC-coated bead studies involving a refolded C2-domain containing a 20-residue affinity tag including 6xHis (*Bittova et al., 1999*; *Stahelin and Cho, 2001b*; *Ward et al., 2012*), weaker affinity of Y96A and D65A mutants for DHPC was observed. In our experiments, the Ca$^{2+}$-concentration-dependence of the process that drives C2-doman point mutants (Y96A, Y96F, and N65D) to PC membranes was assessed by Förster resonance energy transfer (FRET) between Trp71 of the C2-domain and dansyl-PE (1,2-dioleoyl-sn-glycero-3-phosphoethanolamine-N-(5-dimethylamino-1-naphthalenesulfonyl) in the PC model membranes (*Figure 3B*). The need for greater Ca$^{2+}$ concentrations to induce Y96A or N65D partitioning to the PC model membranes compared to either Y96F or control C2-domain partitioning, was clearly evident. *Figure 3C* shows additional FRET data obtained by titration of the C2-domain mutants with increasing amounts of PC model membranes at constant Ca$^{2+}$ concentration (50 µM). The summarized data (*Figure 3D*) confirm the weaker PC-binding affinity of the Y96A and N65D mutants compared to the Y96F mutant or the control C2-domain.

Since the preceding mutational analyses focused on the role of PC-interacting residues in isolated C2-domain on protein binding to PC membranes, the functional impact of these same mutations on the catalytic activity of cPLA$_2\alpha$ (C2-domain + catalytic domain) was also determined using established mixed-micelle assays (*Wijesinghe et al., 2009*). By using surface dilution kinetics and tracking the total mass of arachidonic acid release from 1-palmitoyl- 2-arachidonoyl-sn-glycero-3-phosphocholine (PAPC) via UPLC-MS/MS, protein interfacial partitioning and enzymatic activity were analyzed (*Figure 3E and F*, *Table 2 and 3*). Notably, the data support the functional importance of Tyr96 for PC association. Specifically, the association of the Y96A mutant with PAPC-containing mixed micelles is significantly reduced, that is , the dissociation rate is increased as signified by an increased Ks$^A$, compared to that of either the conservative Y96F point mutant or the control enzyme (*Figure 3E*; *Table 2*). Also, the turnover of the Y96A mutant enzyme displayed allosteric sigmoidal kinetics once the enzyme was bound to the surface of micelles containing PAPC (e.g., a significant difference in V$_{max}$ was observed) without significantly affecting K$_{0.5}$(i.e., the PAPC concentration that produces half-maximal enzyme velocity) (*Figure 3F*; *Table 3*). These data indicate that Tyr96 is key for recognizing and binding PC in the membrane but not for the enzymatic activity of the catalytic domain towards substrate once the enzyme is associated with the membrane.

In cPLA$_2\alpha$, disruption of the N65 interaction with the PC phosphate group by point mutation to Asp (N65D) also significantly reduced the association with micelles containing PAPC when compared to the same interaction for the WT enzyme and the N64A mutant (*Figure 3E*; *Table 2*), in agreement with the SPR and FRET findings for purified C2-domain mutants. Notably, however, the reduced V$_{max}$ observed for N65D cPLA$_2\alpha$, alongside the lack of a significant effect on the enzyme's affinity for substrate, K$_{0.5}$ (*Figure 3F*; *Table 3*), suggests a possible role in enzyme lateral diffusion (i. e., 'scooting') once bound to the membrane. Of note, N64 replacement with Ala (N64A) had no significant effect on either the dissociation rate or the kinetic parameters of the enzyme. Overall, the mutagenesis data strongly support the key role played by both Y96 and N65 in recognizing and binding to PC-rich membranes, with N65 and Y96 regulating PC-binding affinity and playing a

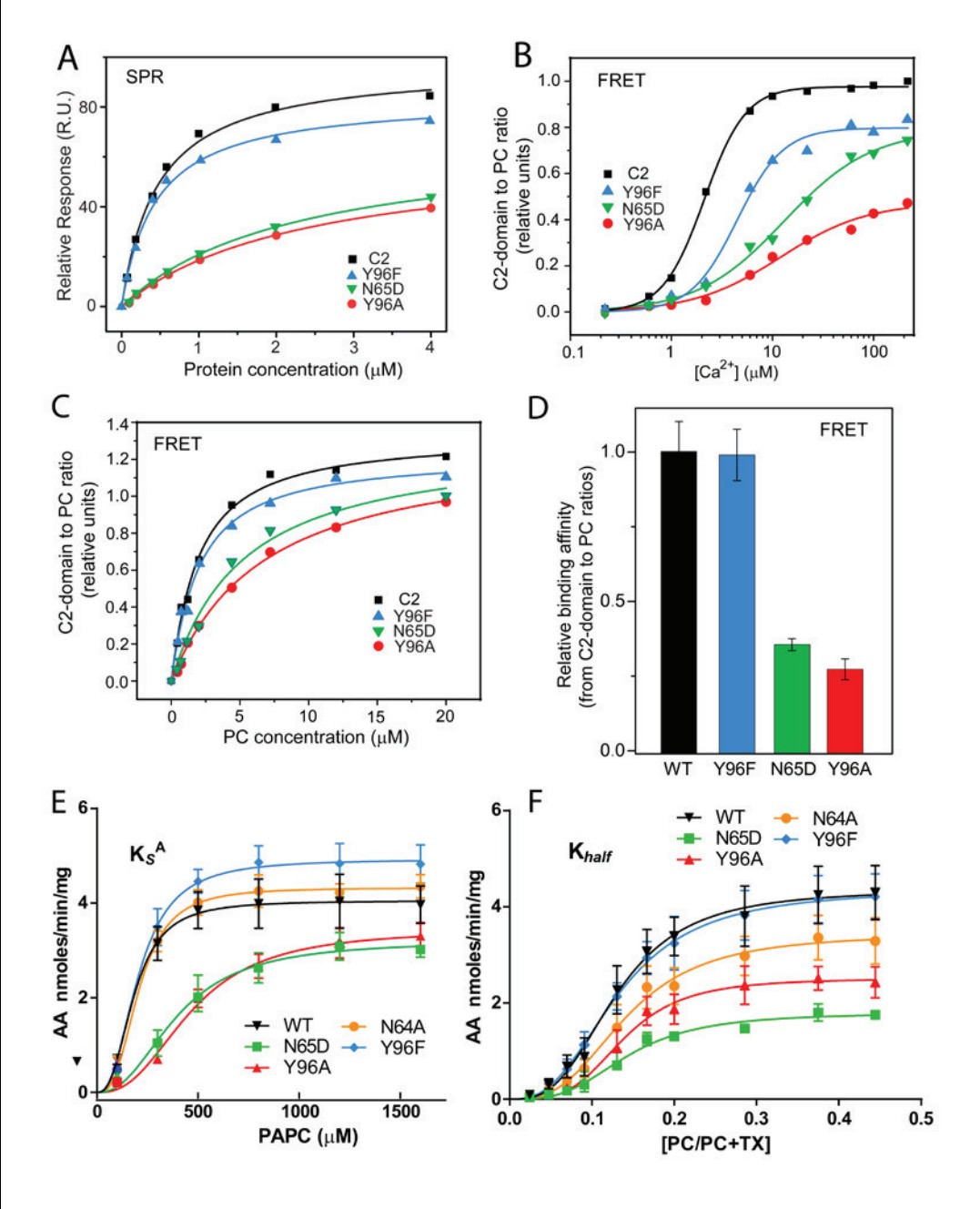

**Figure 3.** Membrane partitioning of cPLA$_2\alpha$ C2-domains and cPLA$_2\alpha$ catalytic activities of point-mutated C2-domains in the PC-binding region. (**A**) SPR binding isotherms showing point mutant and control protein equilibrium adsorption to immobilized 1-palmitoyl-2-oleoyl phosphatidylcholine (POPC) vesicles saturating a L1 sensor chip at 5 µl/min solution flow rates (see 'Materials and methods'). (**B**) FRET binding isotherms showing the Ca$^{2+}$ dependence of point mutant and control protein (0.5 µM) equilibrium adsorption to POPC–DHPC bicelle-dilution vesicles (4 µM) (see 'Materials and methods'). (**C**) FRET-binding isotherms showing the POPC–DHPC bicelle-dilution vesicle dependence of point mutant and control protein (0.5 µM) equilibrium adsorption at 50 µM Ca$^{2+}$ (see 'Materials and methods'). (**D**) Relative binding affinity of C2-domain point mutants and control protein obtained for binding isotherms shown in panel (C). (**E**) Effect of the Y96F, Y96A, N64A, and N65D mutations on the dissociation constant (K$_s^A$) of human cPLA$_2\alpha$ activity. Proteins were purified as described *Stahelin et al. (2007)*. Activity was measured as a function of PC molar concentration for 60 min at 37˚C. The PC mole fraction was held constant at 0.285. cPLA$_2\alpha$ activities (nmol of arachidonic acid released/min/mg of recombinant cPLA$_2\alpha$) were collected on eight separate occasions and are presented as n = 4 for Y96F, n = 4 for Y96A, n = 4 for N64A, n = 4

*Figure 3 continued on next page*

*Figure 3 continued*

for N65D, and n = 8 for WT. Error = standard deviation. $R^2$ values are 0.9021, 09609, 0.9586, 0.9780, and 0.9485 for WT, Y96F, Y96A, N64A, and N65D, respectively. (F) Effect of Y96F, Y96A, N64A, and N65D mutations on the allosteric sigmoidal constant ($K_{half}$) of human cPLA$_2\alpha$ activity. Activity was measured as a function of increasing PC mole fractions for 60 min at 37°C. The PC mole fraction ([PC]/[PC]+[TX-100]) was 0.024 at 50 µM PC, 0.047 at 100 µM PC, 0.069 at 150 µM, 0.091 at 200 µM, 0.13 at 300 µM PC, 0.166 at 400 µM, 0.2 at 500 µM PC, 0.28 at 800 µM PC, 0.37 at 1200 µM PC, and 0.44 at 1600 µM PC. cPLA$_2\alpha$ activities (nmol of arachidonic acid released/min/mg of recombinant cPLA$_2\alpha$) were collected on ten separate occasions and are presented as n = 4 for Y96F, n = 4 for Y96A, n = 4 for N64A, n = 4 for N65D, and n = 4 for WT. Error = standard deviation. $R^2$ values are 0.9413, 0.9577, 0.9407, 0.9376, and 0.9761 for WT, Y96F, Y96A, N64A, and N65D, respectively.

The online version of this article includes the following source data for figure 3:

**Source data 1.** Membrane partitioning data for cPLA$_2\alpha$ C2-domains mutated in PC binding region.
**Source data 2.** Activity data for cPLA$_2\alpha$ C2-domains mutated in PC binding region.

potential role in the enzyme's ability to 'scoot' while bound to the membrane and thus to cleave substrate, as previously modeled by Gelb and co-workers (*Bayburt and Gelb, 1997*).

## Lipid specificity of the cPLA$_2\alpha$ C2-domain

To further evaluate the apparent preference of the cPLA$_2\alpha$ C2-domain for PC, SPR analyses were carried out using phosphoglycerides with different polar headgroups (*Figure 4A*). The relative binding affinity of the C2-domain for POPC was found to be ~5 fold greater than that for chain-matched phosphatidylserine (POPS), phosphatidic acid (POPA), phosphatidylglycerol (POPG), or phosphatidylinositol (POPI) (*Figure 4B*). With phosphatidylethanolamine (POPE), lipid adsorption to the SPR L1 Sensor Chip was quite low compared to that of the other phospholipids. To circumvent this issue, 30 mole% POPE was co-mixed with either POPC or POPS. No significant change was found for C2-domain binding to POPC vesicles containing POPE or POPS compared to binding to pure POPC vesicles, showing the dominating effect of the PC headgroup. Yet, C2-domain binding to the POPS vesicles was slightly improved by 30 mole% POPE (see *Figure 4—figure supplement 1*). Our single phosphoglyceride SPR results agree with previous FRET studies (Trp to dansyl-PE) in which binding to lipid vesicles was assessed using a slightly longer, re-folded, recombinant, human cPLA$_2\alpha$ C2-domain (*Nalefski et al., 1998*).

Because of the proposed preference of the cPLA$_2\alpha$ C2-domain for the PC headgroup, we evaluated C2-domain binding to sphingomyelin (SM), which also has a phosphorylcholine headgroup (*Nalefski et al., 1998*; *Leslie and Channon, 1990*; *Klapisz et al., 2000*; *Nakamura et al., 2010*). An issue in need of clarification is whether the reported inhibition of cPLA$_2\alpha$ activity by SM arises from diminished membrane binding driven by the C2-domain or simply because of the inability of the cPLA$_2\alpha$ esterase (i.e., catalytic domain) to hydrolyze SM after binding to the membrane. Using SPR, we detected significantly weaker binding affinity of C2–domain for vesicles composed of N–oleoyl SM ($K_d = 0.93 \pm 0.28 \times 10^{-5}$) compared to POPC ($K_d = 4.2 \pm 0.8 \times 10^{-7}$), despite the shared phosphorylcholine headgroup and the presence of 50 µM CaCl$_2$ (*Figure 4C*). Notably, SPR measurements involving the Y96F, Y96A, and N65D mutants revealed similar *relative* decreases in binding to 18:1-SM vesicles compared to POPC vesicles (*Figure 4D*). These findings support the key involvement of

**Table 2.** $K_d$ values determined by SPR.

| Protein | $K_d$ (M) | Fold increase[*] |
|---|---|---|
| WT-C2-domain | $(4.2 \pm 0.8) \times 10^{-7}$ | —— |
| Y96F-C2-domain | $(4.3 \pm 0.5) \times 10^{-7}$ | 1 |
| Y96A-C2-domain | $(2.4 \pm 0.4) \times 10^{-6}$ | 5.7 |
| N65D-C2-domain | $(2.2 \pm 0.5) \times 10^{-6}$ | 5.2 |

[*]Fold increase in $K_d$ relative to the C2-domain binding to POPC vesicles. $K_d$ values were determined from the normalized saturation binding responses ($R_{eq}$) at the protein concentrations shown in *Figure 4—figure supplement 1* after fitting by nonlinear least squares analysis using $R_{eq} = R_{max}/(1 + K_d/C)$.

**Table 3.** Kinetic activity parameters for point-mutated PC-binding-site residues in the C2-domain of cPLA$_2\alpha$*.

| Protein | Ks$^A$ (μM) | V$_{max}$ (nmol/min/mg) |
|---|---|---|
| WT-cPLA$_2\alpha$ | 182.8 ± 12.5 | 4.053 ± 0.092 |
| Y96F-cPLA$_2\alpha$ | 205.8 ± 12.7 | 4.930 ± 0.111 |
| Y96A-cPLA$_2\alpha$ | 467.5 ± 31.6 | 3.438 ± 0.177 |
| N65D-cPLA$_2\alpha$ | 394.9 ± 29.8 | 3.203 ± 0.166 |
| N64A-cPLA$_2\alpha$ | 207.7 ± 9.63 | 4.328 ± 0.069 |

*Analyses for data shown in **Figure 3E**.

Tyr96 and Asn65 in binding the phosphorylcholine headgroup of SM but indicate that other factors contribute to the weaker binding of the cPLA$_2\alpha$ C2-domain to SM compared to PC.

It is noteworthy that *N*-oleoyl SM was used for comparison with POPC to avoid effects related to lipid-packing differences of bilayer liquid-crystalline versus gel phase states and to provide relatively well-matched aliphatic chains (18:1-SM vs POPC). Nonetheless, the known propensity of SM for intra- and inter-molecular hydrogen bonding enables more self-interaction than is possible for POPC even when both lipids are in similar bilayer phase states (*Smaby et al., 1996*; *Brown and Brockman, 2007*; *Zhai et al., 2014*; *Slotte, 2016*). The net effect for SM is not only moderately tighter lateral packing in bilayers (compared to POPC) but a phosphorylcholine headgroup with altered conformation and restricted orientational freedom that may partially mitigate interaction with C2-domain binding site residues.

Another important parameter that can affect protein behavior at the membrane interface is the dipole potential that arises from the oriented lipid polar functional groups and associated water molecules located in the interfacial region separating the aqueous phase and hydrocarbon-like interior of the membrane (*Brockman, 1994*; *McIntosh et al., 1992*; *Brockman et al., 2004*). Because of structural differences in the lipid backbones (glycerol with esterified acyl chains in POPC versus sphingosine with an amide-linked acyl chain in SM) (*Figure 1C and 4C*), their dipole potentials differ by ~90 to 240 mV (depending on lipid phase state). This difference has been traced to the similar positioning of the carbonyl group in the esterified fatty acyl chain of POPC and the 3-OH group in the SM sphingoid chain (*McIntosh et al., 1992*). Given the importance of the bilayer dipole potential in the regulation of amphitropic protein and drug translocation to the membrane interface (*Alakoskela et al., 2004*; *Brockman, 1994*; *Kovács et al., 2017*; *Richens et al., 2015*), the lower dipole potential of SM appears to be a probable contributor to the weaker interaction of the cPLA$_2\alpha$ C2-domain for SM bilayers.

The structure of the C2-domain–DHPC complex also reveals the role played by the PC *sn-1* ester linkage and associated carboxyl moiety (*Figure 1C*) interacting with the Ca$^{PC}$ ion. Although the structurally equivalent amide-acyl linkage in SM (*Figure 4C*) could interact with the Ca$^{PC}$ ion, the weaker electronegativity of the carbonyl group in the amide linkage (compared to an ester linkage) may contribute to the diminished binding affinity. Regardless, it is noteworthy that the cPLA$_2\alpha$

**Table 4.** Kinetic activity parameters for point-mutated PC-binding-site residues in the C2-domain of cPLA$_2\alpha$ *

| Protein | K$_{0.5}$ (mole fraction) | V$_{max}$ (nmol/min/mg) |
|---|---|---|
| WT-cPLA$_2\alpha$ | 0.130 ± 0.007 | 4.352 ± 0.183 |
| Y96F-cPLA$_2\alpha$ | 0.132 ± 0.007 | 4.362 ± 0.172 |
| Y96A-cPLA$_2\alpha$ | 0.139 ± 0.006 | 2.510 ± 0.098 |
| N65D-cPLA$_2\alpha$ | 0.143 ± 0.005 | 1.791 ± 0.050 |
| N64A-cPLA$_2\alpha$ | 0.142 ± 0.008 | 3.436 ± 0.168 |

*Analyses for data shown in **Figure 3F**

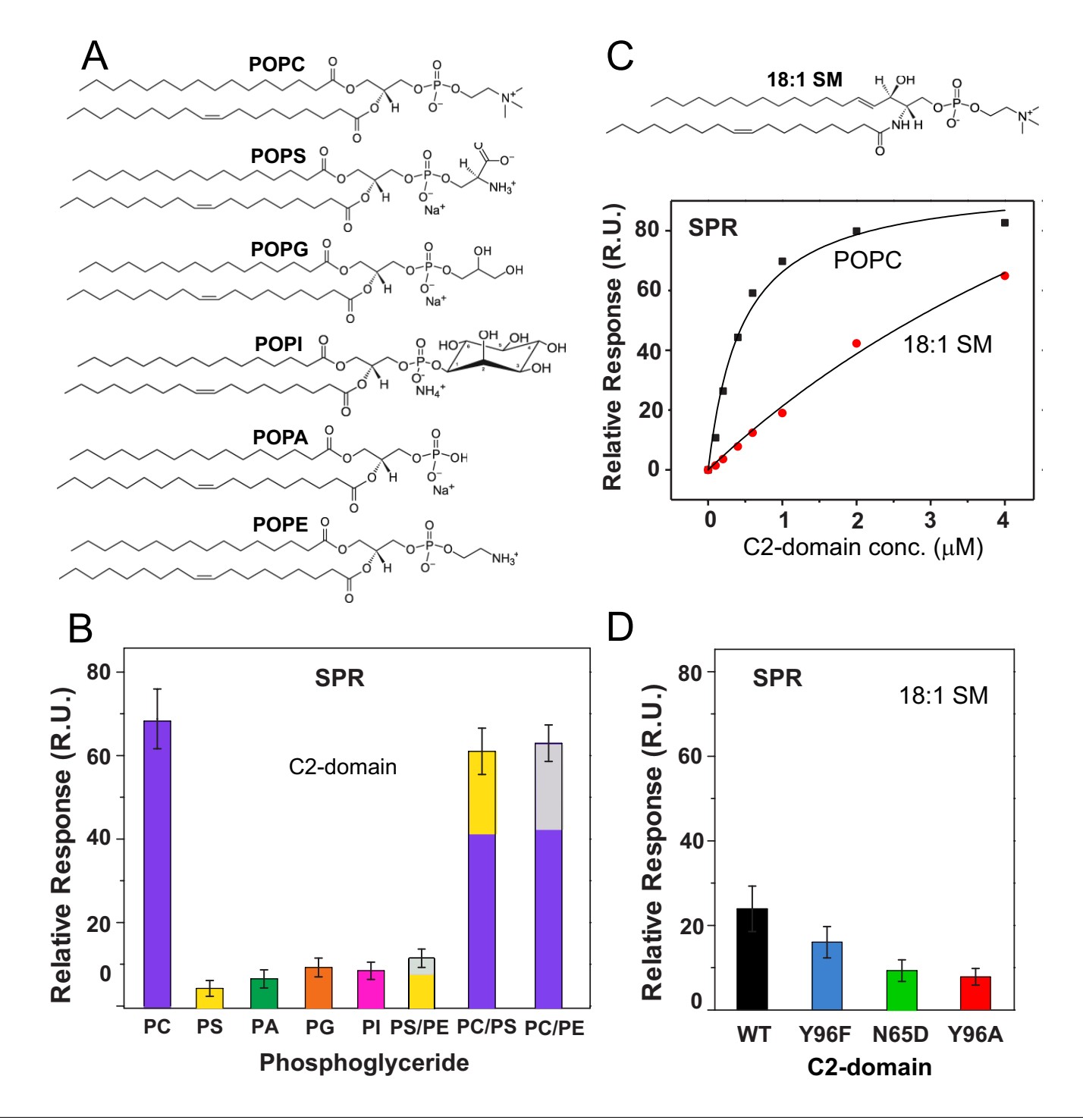

**Figure 4.** cPLA$_2\alpha$ C2-domain binding affinity for phosphoglyceride and sphingomyelin (SM) vesicles. (**A**) Phosphoglyceride structural formulas. (**B**) Relative affinities of the C2-domain (1 μM) for different phosphoglycerides obtained by SPR. Molar ratios for PS/PE, PC/PS and PC/PE mixed composition vesicles are 7:3. (see **Figure 4—figure supplement 1**). (**C**) SPR binding isotherms showing C2-domain equilibrium adsorption to immobilized POPC or 18:1-SM vesicles as a function of protein concentration (see **Figure 4—figure supplement 2** for additional information). (**D**) Effect of C2-domain mutations (1 μM) on binding to 18:1 SM obtained by SPR (see 'Materials and methods' for other details).
The online version of this article includes the following source data and figure supplement(s) for figure 4:

**Source data 1.** cPLA$_2\alpha$ C2-domain binding affinity for phosphoglyceride and sphingomyelin (SM) vesicles.
**Figure supplement 1.** Assessment of C2-domain binding to different phosphoglycerides.
*Figure 4 continued on next page*

eLife Research article

Structural Biology and Molecular Biophysics

*Figure 4 continued*

**Figure supplement 2.** The concentration- and time-dependence of C2-domain adsorption/desorption to/from immobilized POPC and 18:1 SM vesicles measured by SPR.

association with SM-enriched membranes will not enable SM hydrolysis because cPLA$_2\alpha$ is an esterase and its catalytic domain can hydrolyze neither the sphingoid chain nor the amide-linked acyl chain in SM. Our SPR data showing 4-to 5-fold weaker C2-domain binding to SM vesicles than to POPC clarify somewhat conflicting earlier findings regarding the molecular basis for SM inhibition of cPLA$_2\alpha$ action (*Nalefski et al., 1998*; *Nakamura et al., 2010*). Notably, despite the weak affinity of the cPLA$_2\alpha$ C2-domain for SM, in vivo inhibition of activity is unlikely because of intracellular topological factors. cPLA$_2\alpha$ translocates to the Golgi cytosolic face to function; whereas SM is synthesized at the Golgi lumen before being exported to the plasma membrane (*Tafesse et al., 2007*; *Deng et al., 2016*).

## Discussion

C2-domains occur in many (>127) eukaryotic proteins. The C2-domain superfamily includes two families: i) PLC-like variants (including cPLA$_2\alpha$), known as the P-family or type II topology and ii) synaptotagmin (Syt)-like variants referred to as the S-family or type I topology, based on their circularly permuted topologies that generate a different orientation of their eight β-strands (*Corbalan-Garcia and Gómez-Fernández, 2014*). In both C2-domain families, Ca$^{2+}$ ions bind acidic residues within Ca$^{2+}$ binding loops that converge at one end of the β-sandwich structure. The resulting electrostatic neutralization serves as molecular glue to bridge the C2–domain to the phosphoglyceride membrane. Previous analyses of the cPLA$_2\alpha$ C2-domain D93N and N65A mutants, which have defects in the Ca1 and Ca4 sites, respectively, indicate that occupation of the Ca1 site is more essential for membrane binding and activity, perhaps because of favorable conformational changes in the nonpolar residue aliphatic and aromatic side-chains of CBL3 that stabilize membrane interaction (*Bittova et al., 1999*; *Stahelin and Cho, 2001a*). Other mutational studies have shown the key role played by various nonpolar residues located in the CBLs in membrane docking and insertion by the C2-domain upon Ca$^{2+}$ binding and neutralization of nearby anionic Asp residues (*Nalefski et al., 1994*; *Davletov et al., 1998*; *Dessen et al., 1999*; *Perisic et al., 1998*; *Bittova et al., 1999*; *Six and Dennis, 2003*; *Burke et al., 2008*; *Stahelin and Cho, 2001a*; *Stahelin and Cho, 2001b*; *Nalefski and Falke, 1998*; *Ball et al., 1999*; *Malmberg et al., 2003*; *Malmberg and Falke, 2005*; *Málková et al., 2005*). For instance, studies of F35A, M38A, L39A in CBL1 as well as Y96A, V97A, and M98A in CBL3 support their involvement in the interaction with DHPC-coated beads and their importance for optimal cPLA$_2\alpha$ activity (*Bittova et al., 1999*). Several of these earlier biophysical studies have provided detailed insights into general aspects of membrane interaction involving the cPLA$_2\alpha$ C2-domain. Not previously addressed, however, was the possibility that the cPLA$_2\alpha$ C2-domain is structurally designed to target PC-rich membrane regions, thereby helping to increase the enzymatic efficiency of the catalytic domain, which prefers PCs carrying polyunsaturated acyl chains. The current study reveals the molecular basis through which Ca4 and Ca$^{PC}$, as well as Tyr96, Ala94, Asn64, Asn65, and Leu39, work together to target lipids containing phosphorylcholine headgroups (such as PC and SM) whereas other nonpolar residues (Phe35, Met38, Met98, and Val97) appear to promote more nonspecific interactions with the phosphoglyceride bilayer.

The molecular details provided by our structure-function data on DHPC binding with the cPLA$_2\alpha$ C2-domain provide further insight into the C2-domain's penetration of the membrane interface. *Figure 5A* depicts an *ad hoc* model showing the interaction of the C2-domain–DHPC complex with a PC membrane interface, within the context of an earlier docking and penetration model associated with a more general membrane interaction by the cPLA$_2\alpha$ C2-domain (*Figure 5B*). In addition, we consider structural parameters determined for liquid-crystalline (Lα phase) bilayers consisting of dioleoyl PC (*Wiener and White, 1992*). Such parameters include the distances of choline (21.87 Å), phosphate (20.17 Å), and the acyl chain carbonyl groups (15.98 Å) from the bilayer mid-plane, as well as the nitrogen–phosphate distance (4.5 Å) within phosphorylcholine, which orients at 22 ± 4˚ with respect to the bilayer surface. Using the acyl carbonyl groups as markers of the headgroup-hydrocarbon boundary (*Wiener and White, 1992*), the polar headgroup region thickness

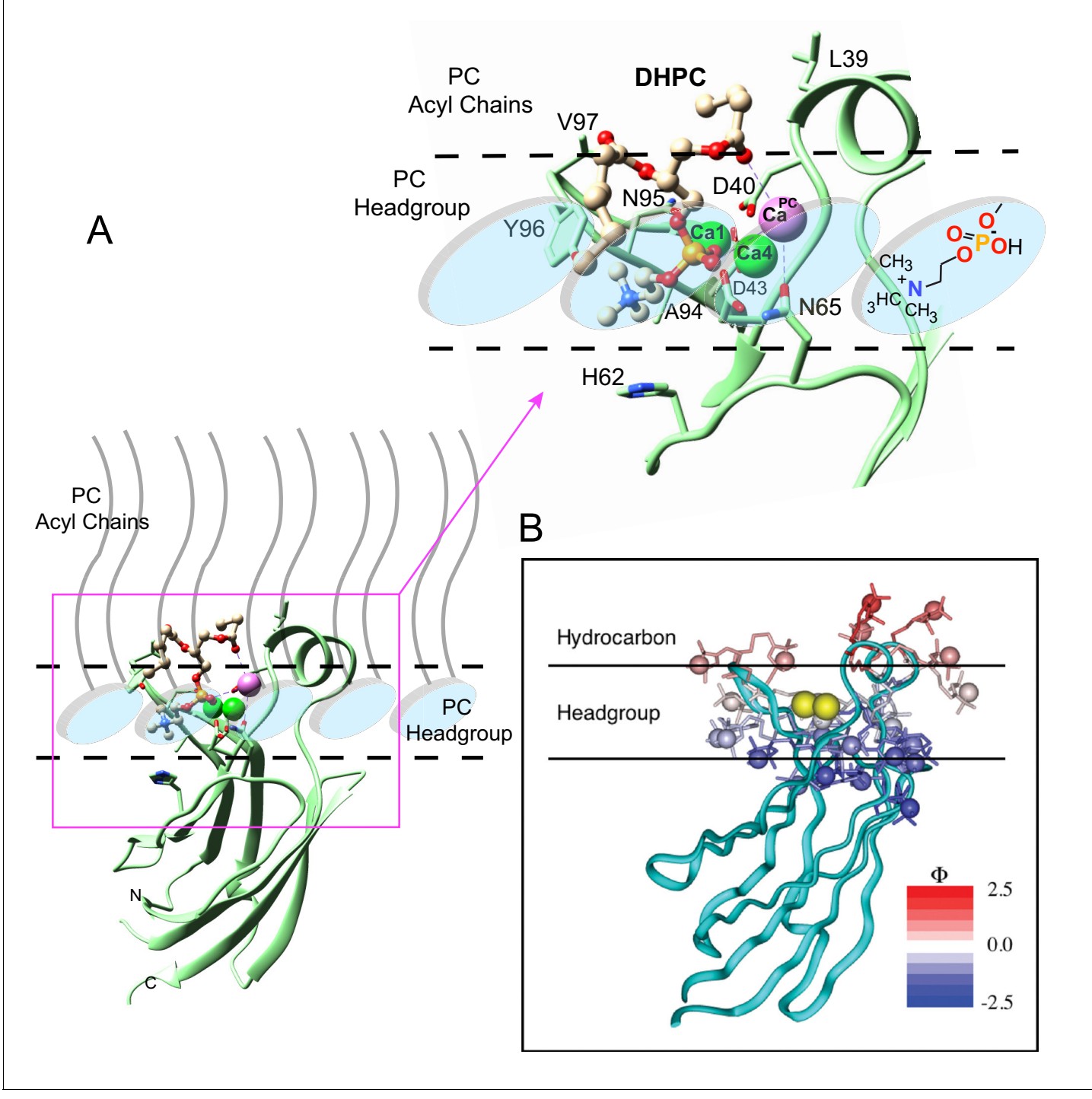

**Figure 5.** Model of the cPLA$_2$α C2-domain selectively interacting with the PC membrane interface. (**A**) Interaction of the C2-domain–DHPC structural complex with a PC membrane interface produced by *ad hoc* modeling. The dashed horizontal lines represent planar boundaries for the lipid headgroup and hydrocarbon regions of the PC bilayer. The crystal structure of the C2-domain–DHPC complex is represented as pale green ribbon. The Ca$^{2+}$ ion that is unique to the C2-domain–DHPC complex is shown as a pale magenta sphere; whereas the other two Ca$^{2+}$ ions are shown as green spheres. In the bound DHPC structure, blue, red, orange, and beige colors represent nitrogen, oxygen, phosphorus, and carbon atoms, respectively. The zoomed view shows how the orientation and position of the PC headgroup bound by the C2-domain requires no major conformational change relative to those of unbound PC headgroups comprising the membrane interface. Membrane docking orientation and penetration depth by the C2-domain are based on previous data illustrated in panel (**B**) (*Nalefski and Falke, 1998*; *Ball et al., 1999*; *Malmberg et al., 2003*; *Malmberg and Falke, 2005*). (**B**) cPLA$_2$α C2-domain docking orientation and penetration depth at the membrane interface, as determined by electron paramagnetic resonance power saturation. [Reprinted (adapted) with permission *Malmberg et al., 2003*, Biochemistry 42, 13227–13240. Copyright: American

*Figure 5 continued on next page*

*Figure 5 continued*
Chemical Society.]. The crystal structure of the lipid-free C2-domain (PDB: RLW) is represented by the cyan ribbon with two $Ca^{2+}$ ions shown as yellow spheres. The horizontal lines represent planar boundaries for the lipid headgroup and hydrocarbon regions of the bilayer. Protein spin labels oriented in their final optimized conformations are colored according to their measured depth parameters (Φ), with positive and negative depth parameters indicated by increasing red and blue color intensity, respectively.

equals ~8.5–9.0 Å (based on ~5.9 Å for the acyl carbonyl group to choline nitrogen, when allowing plus 2.5–3.0 Å for the hydration of choline). In the C2-domain–DHPC crystal complex, slightly larger distances are observed, such as 6.9 Å for *sn*-2 carbonyl to choline nitrogen and 4.7 Å for the nitrogen-to-phosphate distance. Membrane penetration depths estimated from the crystal structure data for CBL3 Val97 and for CBL1 Ile39 are 10–10.5 Å and 12–12.5 Å, respectively, which are comparable with previous data (*Malmberg et al., 2003*) and which represent a ~78–85% penetration depth relative to the mid-plane of a fluid bilayer (*Wiener and White, 1992*).

In other recent studies, $cPLA_2\alpha$ has been implicated as an inducer of membrane structural changes in cells (*Grimmer et al., 2005*; *San Pietro et al., 2009*; *Cai et al., 2012*). This function occurs independently of catalytic activity, but is important for physiological processes such as the regulation of Fc-receptor-mediated phagocytosis (*Zizza et al., 2012*). The membrane structural alterations that are mediated by $cPLA_2\alpha$ have been linked to the C2–domain, which can generate membrane curvature and tubulation in vitro (*Ward et al., 2012*). Penetration by the $cPLA_2\alpha$ C2-domain into POPC or POPC/POPE/POPS membranes induces positive membrane curvature that is abrogated by Y96A mutation, F35A/L39A double mutation, or $Ca^{2+}$ chelation by EGTA (ethylene glycol-bis(β-aminoethyl ether) (*Ward et al., 2012*). In this regard, it is interesting to note that the crystal structure of the C2-domain–DHPC complex contains three C2-domain molecules (protomers A, B, C) in the asymmetric unit, with the six protomers in two asymmetric units forming a ring-like structure (*Figure 1—figure supplement 2*, upper left panel). The DHPC molecules contribute to the molecular packing of neighboring molecules by locating inside the ring, resulting in a tube-like structure that is enclosed by the C2-domain ring (*Figure 1—figure supplement 2*, upper right panel). Thus, the crystal packing superstructure of the $cPLA_2\alpha$ C2-domain complexed with DHPC supports the induction of positive membrane curvature and tubulation reported for this C2-domain (*Ward et al., 2012*), an arrangement not supported well by the different crystal packing structure of lipid-free C2-domain (*Figure 1—figure supplement 2*, lower panel).

## Comparison of phospholipid recognition by the PKCα C2-domain and other C2-domains

Structural analyses of four other C2-domains containing bound phosphoglycerides have been reported. In the C2-domains of PKCα and rabphilin-3A, phosphatidylinositol 4,5-bisphosphate binding occurs at a basic residue cluster without $Ca^{2+}$ involvement (*Guerrero-Valero et al., 2009*; *Zizza et al., 2012*; *Verdaguer et al., 1999*). By contrast, only two other C2–domain structures have to date been found to utilize bound $Ca^{2+}$ to mediate phosphoglyceride binding: PKCα C2–domain complexed with 1,2-dicaproyl-*sn*-phosphatidyl-L-serine (DCPS), a short-chain PS analog (*Verdaguer et al., 1999*) and Syt1 C2B–domain complexed with phosphoserine (*Guillén et al., 2013*). In the DCPS–PKCα C2–domain structure (*Figure 6B*), one of three bound $Ca^{2+}$ ions is coordinated in a position similar to that of Ca1 in the $cPLA_2$ C2–domain (*Figure 6A*), whereas the others are located at different positions far from CBL2 (*Figure 6B*). In sharp contrast to DHPC recognition by $cPLA_2\alpha$, binding of the phosphoryl group of DCPS involves only Ca1. The seryl moiety of the head group docks mainly with CBL1 but in an orientation (*Figure 6B*) that is almost opposite to that of the bound phosphorylcholine in the $cPLA_2\alpha$ C2–domain–DHPC complex (*Figure 6A*). The seryl carboxyl group hydrogen bonds with Asn189 on CBL1, whereas the carbonyl groups of the fatty acid chains interact with CBL2 and CBL3. In the structures of the Syt1 C2B-domain complexed with phosphoserine (*Ferrer-Orta et al., 2017*) (*Figure 6C*) and the lipid-free Syt1 C2A-domain (*Shao et al., 1998*) (*Figure 6—figure supplement 1*), three bound $Ca^{2+}$ ions are coordinated similarly to those in the $cPLA2\alpha$ C2-domain but only the Ca1 position is shared with the $cPLA2\alpha$ C2-domain (*Figure 6A*). In the Syt1 C2B–domain (*Figure 6C*), the seryl moiety docks deeply towards the $Ca^{2+}$ ion sites, whereas the phosphoryl group interacts with Lys366 in CBL3, but not with any $Ca^{2+}$ ions. Within the seryl moiety, the carboxyl group interacts with Ca1 and the main chain of

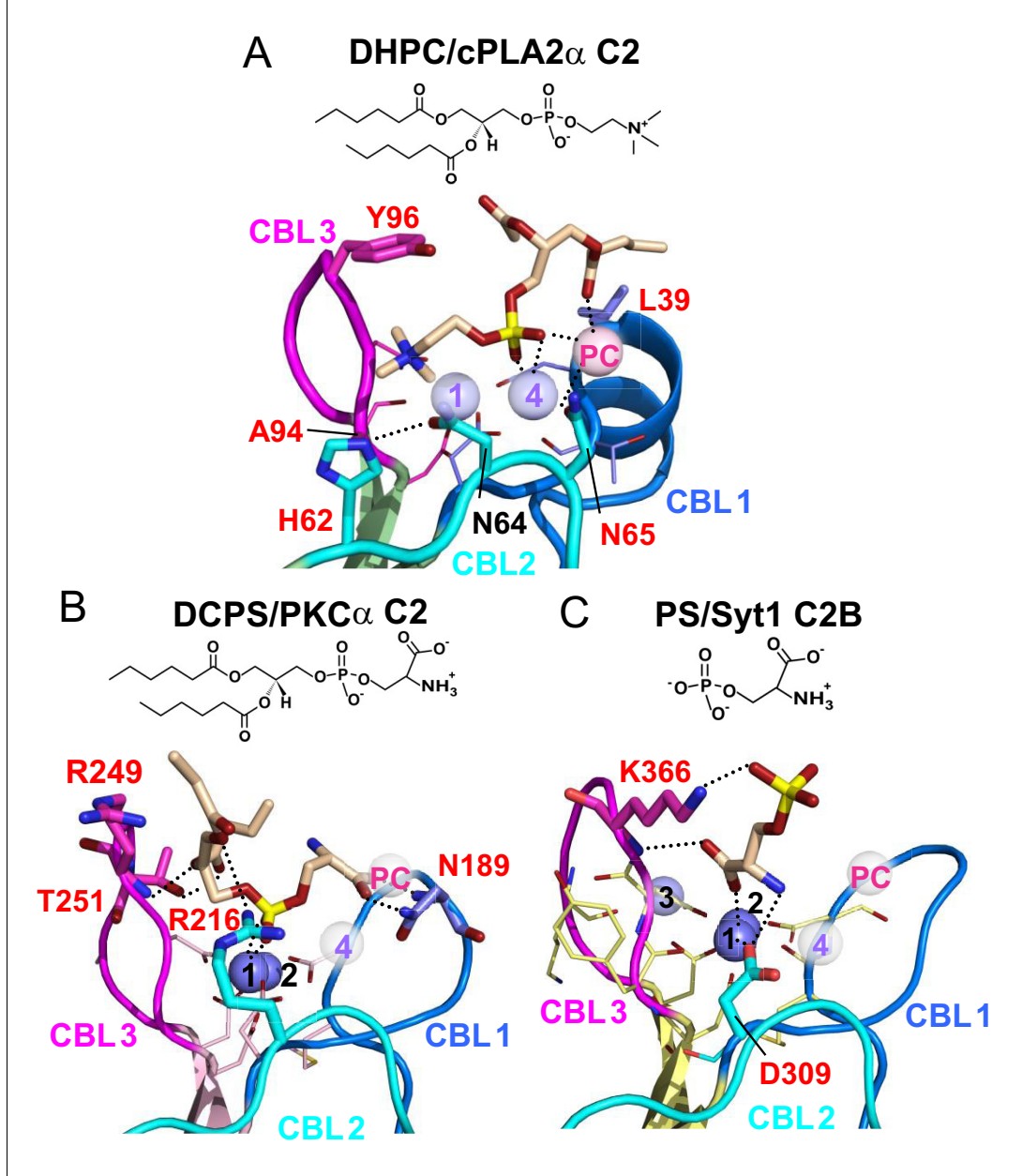

**Figure 6.** Structures of various C2-domains bound to lipids. (A) cPLA$_2\alpha$ C2-domain bound to DHPC determined in this study. (B) PKC$\alpha$ C2-domain bound to phosphatidylserine (PDB 1DSY). (C) Synaptotagmin-1 C2B-domain bound to phosphoserine (PDB 2YOA). For comparison, Ca4 (purple sphere) and Ca$^{PC}$ (pink sphere) in panel (A) are overlaid as pale white spheres in panels (B) and (C). Residues that are interacting directly with ligand are shown as stick models and labeled in red.

The online version of this article includes the following figure supplement(s) for figure 6:

**Figure supplement 1.** Ca$^{2+}$ binding site differences in the synaptotagmin-1 C2A domain versus the cPLA$_2\alpha$ C2-domain complexed with DHPC.

Lys366, while the amine group forms a hydrogen bond with Asp309. This binding mode differs from PS recognition by PKC$\alpha$, although the lack of fatty acyl chains might be the cause of the different interaction. Thus, it is clear that PC recognition by the cPLA$_2\alpha$ C2-domain (*Figure 6A*) is very different from PS recognition by the C2-domains of PKC$\alpha$2 and Syt1 (*Figure 6B and C*). Among eukaryotes, Tyr96 and Asn65 are absolutely conserved in cPLA$_2\alpha$, but not in PKCs or synaptotagmin1 (*Figure 1A*).

Recent structural studies of another cPLA$_2$, cPLA$_2\delta$, have provided molecular insights into the apo-form and the enzyme's catalytic domain complexed with a covalently linked inhibitor (tri-unsaturated 18-carbon phosphonate), but not into phosphoglyceride binding by the enzyme's two tandem C2-domains (*Wang et al., 2016*). Although a general model is proposed for the membrane interaction of cPLA$_2\delta$, involving its tandem C2-domains, this model does not address the issue of phosphoglyceride selectivity.

Our findings point to the key role of cation–π interactions provided by Tyr96 for targeting lipids containing phosphorylcholine headgroups (e.g. PC and SM). Aromatic side-chains (e.g. Tyr and Trp) in mammalian PC transfer protein and in yeast Sec14 transfer protein play a key role in selectivity for PC (*Roderick et al., 2002*; *Kang et al., 2010*; *Schaaf et al., 2008*), as well as in the SM selectivity reported for actinoporin toxins produced by sea anemones (*García-Ortega et al., 2011*). Moreover, Tyr residues that are introduced in close proximity by mutation into peripheral proteins induce a specific interaction with PC in membranes (*Cheng et al., 2013*). Examination of the X-ray structure of the cPLA$_2\alpha$ catalytic domain (*Dessen et al., 1999*) reveals a cluster of aromatic residues (Tyr, Trp, and Phe) in close proximity to the Arg200/Ser228/Asp549 catalytic site residues. Thus, it is tempting to predict that a 'cation-π box' is likely to contribute prominently to the structural underpinning of PAPC selectivity that enables the release of arachidonic acid.

Notably, sequence alignment of the C2-domains of cPLA$_2\alpha$, cPLA$_2\delta$, and three other isoforms shows that the residues (Tyr96 and Asn65), which are so crucial for phosphorylcholine lipid headgroup selectivity by cPLA$_2\alpha$ C2-domain are not conserved in cPLA$_2\delta$ and correspond to Ser97 and Asp66, respectively (*Figure 7*). These residues are unable to undergo cation-π interaction with the -N$^+$(CH$_3$)$_3$ group or favorable polar interaction with the phosphoryl group of the PC headgroup. cPLA$_2\alpha$ is the only cPLA$_2$ isoform that contains a residue (e.g. Tyr96) capable of strong cation-π interaction with the -N$^+$(CH$_3$)$_3$ group in PC. It is also noteworthy that Tyr96 is highly conserved among eukaryotic C2-domains of the cPLA$_2\alpha$ isoform (*Figure 1A* and *Figure 1—figure supplement 1*). These observations suggest that the structure of the C2-domain of cPLA$_2\alpha$ contains design features that promote PC selectivity.

## Conclusions

Our findings support a function for cPLA$_2\alpha$ Tyr96 [via cation-π interaction with -N$^+$(CH$_3$)$_3$] as a key specificity determinant for the phosphorylcholine headgroup of PC, whereas Asn65 tethers with the lipid phosphate moiety and facilitates Ca$^{PC}$ complexation. Notably, Ca4 also functions to tether the PC phosphate moiety to the protein. Bridging by Ca$^{PC}$ to the PC *sn*-2 chain ester linkage further aids enzyme binding and 'scooting' on PC membranes. Further enhancement of C2-domain binding to the membrane, via interaction with Arg59, Arg61 and His 62, is expected when PC-enriched membranes also contain C1P. Taken together, the findings emphasize the unique design features

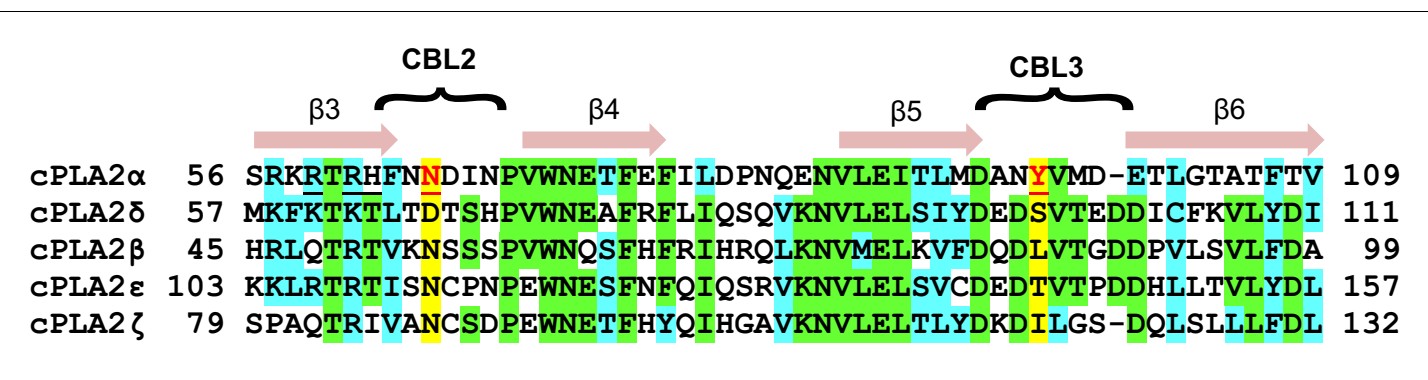

**Figure 7.** C2-domain sequence alignment for five human cPLA$_2$ isoforms showing the uniqueness of the Tyr96 residue in the cPLA$_2\alpha$ C2-domain. β-strand sequences (arrows) as well as CBL2 and CBL3 sequences (bracketed) are shown above the alignment. Green highlights represent identical residues. Cyan highlights represent similar residues. The yellow highlights facilitate comparison of other isoform residues with N65 and Y96 (red) which are key for PC selectivity by the cPLA$_2\alpha$ C2-domain. Underlined black residues in cPLA$_2\alpha$ (R59, R61, and H62) participate in C1P binding (*Stahelin et al., 2007*; *Ward et al., 2013*). Sequence alignment was generated using Clustal Omega.

associated with cPLA$_2\alpha$ C2-domain structure and function, as well as the versatility of lipid recognition exhibited by different C2–domains. In cPLA$_2\alpha$, the C2-domain is structurally designed to target PC-rich membrane regions in order to increase the enzymatic efficiency of the catalytic domain, which prefers polyunsaturated PCs.

# Materials and methods

## Key resources table

| Reagent (species) or resource | Designation | Source or reference | Identifiers | Additional information |
|---|---|---|---|---|
| Strain, strain background (*Escherichia coli*) | BL21 (DE) Star competent cells | ThermoFisher Scientific | SKU# C6010-03 | Cells for protein expression |
| Transfected construct (*E. coli*) | pET SUMO | Snapgene | https://www.snapgene.com/resources/plasmid-files/?set=ta_and_gc_cloning_vectors&plasmid=pET_SUMO_(linearized) | Protein expression vector |
| Commercial assay or kit | JCSG Core Suites | Qiagen | https://www.qiagen.com/us/shop/sample-technologies/protein/crystallization/the-jcsg-core-suites/#orderinginformation | Protein crystallization; crystallization screening kit |
| Chemical compound, drug | 1,2-dihexanoyl-sn-glycero-3-phosphocholine | Avanti Polar Lipids | https://avantilipids.com/product/850305/ | DHPC |
| Chemical compound, drug | 1-palmitoyl-2-oleoyl-glycero-3-phosphocholine | Avanti Polar Lipids | https://avantilipids.com/product/850457/ | POPC |
| Chemical compound, drug | 1-palmitoyl-2-oleoyl-sn-glycero-3-phospho-L-serine | Avanti Polar Lipids | https://avantilipids.com/product/840034/ | POPS |
| Chemical compound, drug | 1-palmitoyl-2-oleoyl-sn-glycero-3-phosphoethanolamine | Avanti Polar Lipids | https://avantilipids.com/product/850757 | POPE |
| Chemical compound, drug | 1-palmitoyl-2-oleoyl-sn-glycero-3-phosphate | Avanti Polar Lipids | https://avantilipids.com/product/840857 | POPA |
| Chemical compound, drug | 1-palmitoyl-2-oleoyl-sn-glycero-3-phospho-(1'-rac-glycerol) | Avanti Polar Lipids | https://avantilipids.com/product/840457 | POPG |
| Chemical compound, drug | N-oleoyl-D-erythro-sphingosyl phosphorylcholine | Avanti Polar Lipids | https://avantilipids.com/product/860587 | 18:1 SM |

*Continued on next page*

*Continued*

| Reagent (species) or resource | Designation | Source or reference | Identifiers | Additional information |
|---|---|---|---|---|
| Chemical compound, drug | 1,2-dioleoyl-sn-glycero-3-phospho ethano lamine-N-(7-nitro-2–1,3-benzoxadiazol-4-yl) | Avanti Polar Lipids | https://avantilipids.com/product/810145 | NBD-PE |
| Software, algorithm | GeneCards | http://genecards.org | RRID:SCR_002773 | Orthologs; retrieval of protein sequences for human, mouse, and chicken proteins |
| Software, algorithm | UniProtKB | http://www.uniprot.org/help/uniprotkb | RRID:SCR_004426 | C2-domain sequences for various proteins and organisms |
| Software, algorithm | NCBI Protein | http://www.ncbi.nlm.nih.gov/protein | RRID:SCR_003257 | Protein sequences for human, mouse, and chicken proteins |
| Software, algorithm | Clustal Omega | http://www.ebi.ac.uk/Tools/msa/clustalo/ | RRID:SCR_001591 | Software package for multiple sequence alignment |
| Software, algorithm | Clustal W2 | http://www.ebi.ac.uk/Tools/msa/clustalw2/ | RRID:SCR_002909 | Multiple sequence alignment program for DNA or proteins. |
| Software, algorithm | UCSF Chimera | http://plato.cgl.ucsf.edu/chimera/ | RRID:SCR_004097 | Program for interactive visualization and analysis of molecular structures |
| Software, algorithm | Protein Data Bank (PDB) | http://www.wwpdb.org/ | RRID:SCR_006555 | Macromolecular structure archive that oversees and reviews deposition and processing data |
| Software, algorithm | Coot | http://www2.mrc-lmb.cam.ac.uk/personal/pemsley/coot/ | RRID:SCR_014222 | Software for macromolecular model building, completion and validation, and protein modeling using X-ray data |
| Software, algorithm | PHENIX | https://www.phenix-online.org/ | RRID:SCR_014224 | Python-based software suite for determination of X-ray crystallographic molecular structures |
| Software, algorithm | PyMol | http://www.pymol.org/ | RRID:SCR_000305 | Data processing, 3D visualization and rendering software |
| Software, algorithm | PDBeFold | http://pdbe.org/fold/ | RRID:SCR_004312 | Co-alignment of compared structures |
| Peptide, recombinant protein | Cytosolic phospholipase A2 | https://www.uniprot.org/uniprot/P47712 | | Human cPLA2 sequence |

*Continued on next page*

*Continued*

| Reagent (species) or resource | Designation | Source or reference | Identifiers | Additional information |
|---|---|---|---|---|
| Peptide, recombinant protein | Cytosolic phospholipase A2 | https://www.uniprot .org/uniprot/P47713 | | Mouse cPLA2 sequence |
| Peptide, recombinant protein | Cytosolic phospholipase A2 | https://www.uniprot .org/uniprot/P49147 | | Chicken cPLA2 sequence |

## Strategy employed to achieve a soluble cPLA2α C2-domain

The C2-domain sequence used to achieve solubility during expression and to avoid the need for protein refolding was developed by consideration of the original lipid-free, $Ca^{2+}$-bound crystal structure (*Perisic et al., 1998*) that used a truncated C2-domain (residues 17–141) from human cPLA2α. However, we did not introduce the C-terminal C139A and C141S substitutions that were implemented to eliminate possible refolding complications induced by Cys residues, because our goal was to recover soluble protein after expression in *Escherichia coli*. In our constructs, the *Bam*HI and *Sal*I restriction sites were used for open reading frame (ORF) ligation into a modified pET-28-SUMO vector (kanamycin-resistance). Prior to insertion, the open reading frames were mutated to remove a single *Bam*HI restriction site within the C2-domain ORF without changing the protein sequence. The complete protein sequences for cPLA2α from human, mouse, and chicken are provided in *Figure 1—figure supplement 1*. The constructs, which were verified by DNA sequencing, enabled the expression of proteins containing Ulp1-cleavable, N-terminal 6xHis-SUMO tags. Testing for their solubility and SUMO-tag cleavability revealed the following:

### Human

- C2-domain$^{1-140}$ or C2-domain$^{15-140}$ expression resulted in mostly insoluble protein located in inclusion bodies.
- C2-domain$^{17-140}$ expression resulted in soluble protein but with a SUMO tag that was inaccessible to cleavage.
- Expression of C2-domain$^{17-140}$ with an inserted Gly or Met residue at the N-terminus (C2-domain $^{Gly/Met-17-140}$) resulted in good expression of soluble protein and cleavage of the SUMO tag.

### Chicken

- Expression of C2-domain$^{16-140}$ resulted in good expression of soluble protein and cleavage of the SUMO tag.

## Protein expression and purification

Protein expression was performed at 20°C in Luria-Bertani medium containing 0.1 mM isopropyl-β-D-thiogalactopyranoside. Cells expressing the C2-domain were harvested, suspended in 20 mM Tris-HCl buffer (pH 8.0) containing 500 mM NaCl and disrupted by French press. After ultracentrifugation, the supernatant was applied onto a Ni-NTA resin (Qiagen) and treated with Ulp1 to remove the N-terminal His6-SUMO tag. Eluted proteins were further purified by anion exchange (HiTrap Q HP, GE Healthcare) and gel filtration (Superdex 75 pg, GE Healthcare) chromatography. Purified protein was concentrated up to 20 mg/mL in 20 mM MES-NaOH buffer (pH 6.0) containing 100 mM NaCl and 2.5 mM $CaCl_2$ and stored at −80°C until use.

## Crystallization and data collection

All lipids used in this study were obtained from Avanti Polar Lipids and dissolved in ethanol. Crystallization conditions were initially screened using a Mosquito crystallization robot (TTP Labtech) with commercial crystallization solution kits, JCSG Core Suite I-IV and PACT Suite (QIAGEN). Despite extensive crystallization trials with human C2-domain$^{Gly-17-140}$, the only resulting protein crystals

contained two bound $Ca^{2+}$ but no bound lipid. Successful crystallization of C2-domain containing bound DHPC and three bound $Ca^{2+}$ions was obtained with the chicken C2-domain[16–140]. The best crystal complexes were obtained from solutions containing 1 mM protein, 5 mM DHPC and reservoir solution containing 100 mM HEPES-NaOH buffer (pH 7.0), 1.4 M $MgCl_2$ and 0.6 M NaCl at 20˚C. Crystal complexes were transferred into a cryoprotective solution containing saturated NaCl and flash-cooled at 100 K. X-ray diffraction data were collected at 100 K on 24-ID-C beamline at the Advanced Photon Source. Data were processed and scaled using HKL-2000 (*Otwinowski and Minor, 1997*). The crystal data and refinement statistics are summarized in *Table 5* and are deposited in the Protein Data Bank (accession code 6IEJ).

## Structure determination and refinement

The structure of the crystal complex was determined by a molecular replacement method using the lipid-free structure of the human $cPLA_2\alpha$ C2-domain (PDB: 1RLW) as a starting model. The built model was refined using alternating cycles of the Coot (*Emsley and Cowtan, 2004*) and PHENIX programs (*Adams et al., 2002*). The model was refined to 2.2 Å resolution. Refinement statistics are summarized in *Table 5*.

**Table 5.** X-ray data collection statistics.

| | Native |
|---|---|
| Data collection | |
| Space group | C222 |
| Cell dimensions | |
| a, b, c (Å) | 108.3, 187.4, 68.8 |
| Wavelength (Å) | 1.00000 |
| Resolution (Å) * | 50–2.20 (2.24–2.20) |
| $R_{sym}$* | 5.9 (36.3) |
| $I/\sigma I$* | 30.9 (1.9) |
| Completeness (%)* | 99.5 (97.7) |
| Redundancy* | 7.7 (6.6) |
| Refinement | |
| Resolution (Å) | 47–2.2 |
| No. reflections | 35,185 |
| $R_{work}$/$R_{free}$ (%) | 22.4/24.9 |
| No. atoms | |
| Protein | 2998 |
| Water | 82 |
| Ion | 11 |
| Ligand | 75 |
| B-factor (Å$^2$) | |
| Protein | 59.9 |
| Water | 56.6 |
| Ion | 54.2 |
| Ligand | 81.3 |
| R.m.s. deviations | |
| Bond lengths (Å) | 0.008 |
| Bond angles (˚) | 1.117 |

One crystal was used for each data set.

*Highest resolution shell is shown in parenthesis.

Electron density mapping of the lipid ligand involved consideration of the chemical structures of both DHPC and MES because MES was included in our crystallization buffer. The omit map (*Figure 1D*) clearly traced density corresponding to the -N$^+$(CH$_3$)$_3$ and phosphate groups of the DHPC head group, as well as extra density corresponding to the ester groups of fatty acid chains rather than to the morpholino and sulfate groups of MES. When we tried to place the MES molecule at this position, the extra strong density corresponding to fatty acid chains stood out. Thus, we identified our structure as DHPC bound to C2–domain.

## Structure and sequence comparison

Multiple sequence alignment was performed by CLUSTALW (*Emsley and Cowtan, 2004*). Pairwise structural comparisons were performed using C$_\alpha$-atom positions by the PDBeFold (*Adams et al., 2002*) in conjunction with SSM (*Krissinel and Henrick, 2004*) and structure figures were prepared using the PyMOL Molecular Graphics System, Version 1.7 Schrödinger, LLC (http://www.pymol.org/) and UCSF Chimera 1.11.2 (http://www.cgl.ucsf.edu/chimera/).

## Point-mutant analyses of C2-domain translocation to PC model membranes

Partitioning of C2-domain and various point mutants to PC model membranes was monitored by FRET and SPR. FRET measurements were performed using Trp/Tyr emission of C2-domain as energy donor and dansyl-PE-POPC-DHPC (5:45:50) bicelle-dilution vesicles as energy acceptors. Bicelle-dilution vesicles were formed by mixing the POPC, dansyl-PE and DHPC in chloroform, drying under a stream of nitrogen and placing under vacuum for ~2 hr, before resuspending in buffer (20 mM Tris, pH 7.5, 150 mM NaCl and 50 μM CaCl$_2$). Unilamellar POPC vesicle preparation by POPC/DHPC bicelle mix dilution is detailed in *Gao et al., 2020*; *Gao et al., 2021*. Binding reactions included C2-domain (0.5 μM) and various amounts of bicelle-dilution vesicles (PC concentration 0.44 to 20 μM) in 2.5 ml of buffer. In binding reactions assessing calcium dependence (2.5 ml total volume), the protein and bicelle dilution-vesicle concentrations (0.5 μM and 4 μM, respectively) were held constant while the Ca$^{2+}$ was varied. FRET measurements were performed at 25°C in a temperature-controled (±0.1°C) cuvette (NesLab RTE-111, ThermoFisher) using a SPEX FluoroLog-3 spectrofluorimeter (Horiba Scientific). Excitation and emission wavelengths were 284 nm and 520 nm, with band-pass settings of 5 and 10 nm, respectively. Inner filter effects were avoided by using low protein concentration (optical density @ 295 nm <0.1). Relative FRET was calculated as (I$_{obs}$ − I$_{min}$) / (I$_{max}$ − I$_{min}$), where I$_{min}$ is the dansyl emission in the absence of Ca$^{2+}$ and I$_{max}$ is the maximal energy transfer obtained from the binding curve. FRET fluorescence data were plotted as the relative fluorescence signal versus PC concentration and fitted to the equation described by *Rao et al. (2005)*.

SPR measurements were performed using a Biacore T200 system (GE Healthcare Bio-Sciences Corp) at 25°C under previously described conditions (*Stahelin et al., 2007*; *Ward et al., 2012*; *Ward et al., 2013*; *Zhai et al., 2017*; *Ochoa-Lizarralde et al., 2018*). An uncoated flow channel was used as a control surface. POPC vesicles (1 mM), prepared by sonication and centrifugation, were captured to a final surface density of 4000–6000 response units on a L1 Sensor Chip to establish the baseline. Each lipid layer was stabilized by injecting 10 μl of 50 mM NaOH three times. Then 100 μl of protein in 10 mM HEPES (pH 7.4), 0.16 M KCl and 50 μM CaCl$_2$ was injected at 5 μl/min flow rate and protein adsorption was monitored. After 20 min, a switch to buffer lacking protein occurred (*Figure 4—figure supplement 2*) but the strong adsorption of the C2-domain required washing with 10 μl of 50 mM NaOH to regenerate the lipid surface. Complete cleaning of the sensor chip could be accomplished by washing with 20 mM CHAPS detergent. The normalized saturation response R$_{eq}$ was plotted versus protein concentration (C), and K$_d$ values were determined by non-linear least-squares fitting using the equation: R$_{eq}$ = R$_{max}$/(1 + K$_d$/C). Each data set was repeated three times to calculate a standard deviation value.

## cPLA$_2$ activity measurements

A mixed micelle was utilized to measure PAPC hydrolysis by cPLA$_2$ as previously reported (*Wijesinghe et al., 2009*). Briefly 4x assay buffer (2 ml) was made using 320 mM HEPES, 600 mM NaCl, 19.42 μM CaCl$_2$ (10 μM free Ca$^{2+}$), 10 μM EGTA, 4 mM DTT, and 2.39% H$_2$O. For K$_{0.5}$ analysis, micelles were created by drying down ten separate concentrations of PAPC by nitrogen and

then reconstituting in 8 mM Triton-X100. Micelle concentrations are as follows: 50 µM PC (2.4% mol PC), 100 µM PC (4.7% mol PC), 150 µM PC (6.9% mol PC), 200 µM PC (9.1% mol PC), 300 µM PC (13% mol PC), 400 µM PC (16.6% mol PC), 500 µM PC (20% mol PC), 800 µM PC (28.6% mol PC), 1200 µM PC (37.5% mol PC), and 1600 µM PC (44.4% mol PC). For $K_sA$ analysis, all six concentrations of micelles were created by first making a 1600 µM PC micelle by drying down PAPC and reconstituting in Triton-X100. The 1200 µM, 800 µM, 500 µM, 300 µM, and 100 µM PC micelles were made via serial dilution of the 1600 µM micelle with LCMS-grade $H_2O$.

To prepare for enzyme for assays, $cPLA_2\alpha$ (250 ng) was mixed with 30% glycerol and 80 mM HEPES. Assay buffer (25 µL) was then combined with micelles (25 µL), followed by protein mix (50 µL) to produce a final reaction volume of 100 µL. A total of five reactions were prepared for each of the ten concentrations. Immediately following enzyme addition to the micelle – assay buffer mixture, a timer was started and mixtures were placed into a 37°C bead bath. At 15, 30, 45, and 60 min, a 100 µL aliquot was pipetted from each concentration into 500 µL of MeOH containing 10 ng of arachidonic acid-$d_8$ (AAd8) to quench the reaction. Samples then were assessed for PAPC hydrolysis via UPLC-LC/MS. Kinetic and statistical analyses were performed using GraphPad Prism 6 (GraphPad Software Inc).

AA and AAd8 were purchased from Cayman Chemicals and analyzed using an adapted method from our previous report (https://www.ncbi.nlm.nih.gov/pmc/articles/PMC3951269/). AA and AAd8 were separated with an Acentis Express C18 HPLC Column 10 cm x 2.1 mm, 2.7 µm via UPLC using a Shimadzu 2-D UPLC Nexera System in conjunction with a QTRAP 5500 Mass Spectrometer (AbSciex). Mass spectrometry parameters were: Polarity, Negative; Ion Source, Electrospray; Q1 Resolution, Unit; Q3 Resolution, Unit; Curtain Gas, 30; Collision Gas, Medium; IonSpray Voltage, −4500; Temperature, 500; Ion Source Gas 1, 40; Ion Source Gas 2, 60; Entrance Potential, −13. MRM transitions with corresponding declustering potentials (DP), collision energies (CE), and collision cell exit potentials (CXP) were: AA Q1 Mass (da), 303.2; Q3 Mass (da), 259.2; DP, −150 volts; CE, −17 volts; CXP, −14 volts. AAd8 Q1 Mass (da), 311.2; Q3 Mass (da), 267.3; DP, −150 volts; CE, −18 volts; and CXP, −16 volts.

UPLC conditions were: Pumping Mode, Binary Flow; Total Flow, 0.7000 mL/min; Injection Volume, 10 µL; Column Oven, 60 °C. The solvents used for reverse phase UPLC separation across a 6 min run were: Solvent A – 60:40 acetonitrile/water with 0.1% formic acid and 10 mM ammonium formate; and Solvent B – 10:90 acetonitrile/isopropanol with 0.1% formic acid and 10 mM ammonium formate. Solvent conditions for UPLC separation were: 10% Solvent B from 0 to 1 min, linear increase from 10–100% Solvent B from 1 to 4 min; constant 100% Solvent B from 4 to 5 min; at 5 min, a drop from 100–10% solvent B; from 5 to 6 min, 10% constant solvent B; and at 6.1 min, Controller Stop. Because membranes are allosteric activators of $cPLA_2\alpha$ *in vitro* (*Mouchlis et al., 2015*), both kinetic curves were fit using a non-linear regression allosteric sigmoidal best-fit approach. Statistical analyses were a single ANOVA with a Tukey HSD post-hoc test with $p < 0.01$ considered significant.

## Accession number
The atomic coordinates and structure factors for chicken $cPLA_2\alpha$ C2-domain bound to DHPC are deposited in Protein Data Bank under accession code 6IEJ.

## Acknowledgements

This work is based upon research conducted at the Northeastern Collaborative Access Team beamlines, which are funded by the National Institute of General Medical Sciences from the National Institutes of Health (P30 GM124165). The Pilatus 6M detector on 24-ID-C beamline is funded by a NIH-ORIP HEI grant (S10 RR029205). This research used resources of the Advanced Photon Source, a U. S. Department of Energy (DOE) Office of Science User Facility operated for the DOE Office of Science by Argonne National Laboratory under Contract No. DE-AC02-06CH11357. This work was also supported by the Program for Promoting the Enhancement of Research Universities from the Ministry of Education, Culture, Sports, Science and Technology (MEXT), Japan (YH); research grants from the National Institutes of Health via HL125353 (CEC, REB, and DJP), HD087198 (CEC), and RR031535 (CEC); the Veteran's Administration [VA Merit Review, I BX001792 (CEC)]; a Research Career Scientist Award 13F-RCS-002 (CEC); a Memorial Sloan-Kettering Cancer Center Core Grant

P30 CA008748 (DJP); the Maloris Foundation (DJP); and the Hormel Foundation (REB). The contents of this manuscript do not represent the views of the Department of Veterans Affairs or the United States Government.

## Additional information

### Funding

| Funder | Grant reference number | Author |
|---|---|---|
| Ministry of Education, Culture, Sports, Science, and Technology | | Yoshinori Hirano |
| National Institutes of Health | HL125353 (multi-PI) | Rhoderick E Brown<br>Charles E Chalfant<br>Dinshaw J Patel |
| U. S. Department of Veteran Affairs | I BX001792 | Charles E Chalfant |

The funders had no role in study design, data collection and interpretation, or the decision to submit the work for publication.

### Author contributions

Yoshinori Hirano, Conceptualization, Data curation, Formal analysis, Validation, Investigation, Visualization, Methodology, Writing—original draft, Writing—review and editing; Yong-Guang Gao, Conceptualization, Data curation, Formal analysis, Investigation, Methodology, Writing—original draft; Daniel J Stephenson, Formal analysis, Validation, Investigation, Methodology, Writing—original draft; Ngoc T Vu, Formal analysis, Investigation, Methodology; Lucy Malinina, Data curation, Formal analysis, Validation; Dhirendra K Simanshu, Investigation, Methodology; Charles E Chalfant, Conceptualization, Formal analysis, Supervision, Funding acquisition, Validation, Investigation, Writing—original draft, Project administration, Writing—review and editing; Dinshaw J Patel, Conceptualization, Funding acquisition, Validation, Investigation, Project administration, Writing—review and editing; Rhoderick E Brown, Conceptualization, Supervision, Funding acquisition, Investigation, Project administration, Writing—review and editing

### Author ORCIDs

Yoshinori Hirano (iD) https://orcid.org/0000-0001-9888-1616
Yong-Guang Gao (iD) http://orcid.org/0000-0002-9359-4252
Daniel J Stephenson (iD) http://orcid.org/0000-0002-5698-3400
Lucy Malinina (iD) http://orcid.org/0000-0001-7973-1831
Dhirendra K Simanshu (iD) https://orcid.org/0000-0002-9717-4618
Charles E Chalfant (iD) https://orcid.org/0000-0002-5844-5235
Rhoderick E Brown (iD) https://orcid.org/0000-0002-7337-3604

### Decision letter and Author response

Decision letter https://doi.org/10.7554/eLife.44760.sa1
Author response https://doi.org/10.7554/eLife.44760.sa2

## Additional files

### Supplementary files

- Transparent reporting form

### Data availability

Diffraction data have been deposited in PDB under the accession code 6IEJ.

The following dataset was generated:

| Author(s) | Year | Dataset title | Dataset URL | Database and Identifier |
|---|---|---|---|---|
| Yoshinori Hirano, Yong-Guang Gao, Daniel J Stephenson, Ngoc T Vu, Lucy Malinina, Charles E Chalfant, Dinshaw J Patel, Rhoderick E Brown | 2019 | Structural basis of phosphatidylcholine recognition by the C2-domain of cytosolic phospholipase A2$\alpha$ | http://www.rcsb.org/structure/6IEJ | Protein Data Bank, 6IEJ |

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
