## [Decision Letter]

Thank you for submitting your article "Structural basis of phosphatidylcholine recognition by the C2-domain of cytosolic phospholipase A_2_α" for consideration by *eLife*. Your article has been reviewed by three peer reviewers, and the evaluation has been overseen by a Reviewing Editor and Cynthia Wolberger as the Senior Editor. The reviewers have opted to remain anonymous.

The reviewers have discussed the reviews with one another and the Reviewing Editor has drafted this decision to help you prepare a revised submission.

Summary:

This study reports structural data which explains the selective binding of phosphatidylcholine (PC) to the C2 domain of cytosolic phospholipase A_2_α (cPLA_2_). The data reveal the positioning of the bound PC headgroup and identify a novel calcium bridge between cPLA_2_'s C2 and PC's phosphoryl group. This study improves our understanding of how calcium and lipid coordination can be mediated in different ways in C2 domains of different proteins. The reviewers all agreed that the data were of high quality and that the findings were significant, but also raised some points that need to be addressed.

Essential revisions:

1) Please address the following questions about PC selectivity:

The positively charged choline headgroup of PC appears to lie near the positive calcium charge of Ca1. Is it possibly more favorable to have anionic lipid here (PS due to size or even PE due to size/shape)? The authors discuss this in Figure 4—figure supplement 1 by showing different lipid structures and discussing why they may or may not bind compared to PC.

But, could the new structure here actually be revealing something more? Since PC has been studied as the bulk lipid that binds and PC was used to crystallize, perhaps the field has been biased against lipid mixtures (with small anionic lipid such as PS or other in the vesicles)?

The authors suggest PS will have some repulsions due to the anionic headgroup but it is difficult to see this. Also, the phosphoryl group of anionic lipid could be stabilized by the calcium bound to the C2 domain.

Similarly, the PC headgroup is close to His62, which under some conditions could be partially protonated (~pKa near 6).

2) As written, the reduction in binding and/or activity by Y96A or N65D doesn't really prove they are involved in PC selectivity, it just demonstrated mutations reduce affinity and activity. To prove/demonstrate selectivity, one would have to compare the mutants and WT to lipids under these different conditions. The authors may want to consider rephrasing/rewriting these statements based upon their findings and that of the previous literature. For instance, although SM binding is low, what would the effects of mutations be? Would there be selective reduction in PC binding vs. that of SM?

Also, the SM vs. PC comparison is good and the authors are right that it hasn't been addressed much before. That being said, is SM relevant for Golgi or ER targeting of cPLA_2_ vs. that of PC or other lipids?

3) Please address the following points regarding previous work in the field:

The Introduction (paragraph three) points out that the lack of a lipid-bound crystal form of the C2 domain has hampered understanding of the structural basis of lipid activation mechanisms. While this current study does provide the lipid-bound structure to advance the field, it may be worth pointing out how the work of Ed Dennis and colleagues (among others) using molecular dynamics and hydrogen-deuterium exchange mass spectrometry has helped the field in understanding cPLA_2_ activity.

More details should be incorporated on the work done on Y96A and N65A by Bittova et al., 1999, as well as the role of N65A in membrane binding and penetration by Stahelin and Cho, 2001. and work by Burke et al. 2008. Bittova et al. studied the Y96A effects in full-length cPLA_2_ demonstrating a reduction in affinity for DHPC and a reduction in activity, which is confirmed in the current study. Similarly, it was shown that N65A reduced cPLA_2_ DHPC affinity and enzyme activity.

4) Were the studies with full-length protein pursued with rat, human or chicken cPLA_2_? If not chicken, can the C2 chicken structure and lipid binding be directly compared to its influence on the C2-catalytic role in rat or human? If full-length chicken cPLA_2_ was used, how does its catalytic domain sequence compare to rat and human, and how does its activity compare to those published previously for the rat or human enzymes?

Classically, the C2 domain of cPLA_2_ (rat) needed to be refolded from *E. coli* expression due to solubility issues. The authors have made great strides here in expressing the C2 domain from chicken cPLA_2_ in soluble form. This could be of interest to the field so a bit more detail on this solubility in the Materials and methods would be useful.

It would be helpful to have the alignment of the full cPLA_2_ of chicken with rat and human species.

5) In Figure 5, where the authors model the crystal structure onto the schematic drawing and compare to EPR results, it would be more informative if distances are compared. For example, distances from x-ray structure, headgroup dimensions, EPR data can be compared. How deep in the membrane does the C2 domain go from experimental studies?

6) The binding of DHPC in one site is well appreciated but it is expected that the C2 domain will make contact and interact (at least non-specifically) with a number of lipid head groups around CBLs 1 and 3. Can this briefly be incorporated into the discussion model?

Also, in the crystal lattice, do the hydrophobic side chains of the lipid molecule interact with each other?

7) Synaptotagmin 1 C2A domain binds to three calcium ions. Can the authors comment and compare the structures?

8) In the Materials and methods, the authors mention C1P purchased from Avanti but it is not clear when or how they were used in the paper? In crystallization attempts?

---

## [Author Response]

Essential revisions:1) Please address the following questions about PC selectivity:The positively charged choline headgroup of PC appears to lie near the positive calcium charge of Ca1.

Admittedly, the viewing angles presented for the C2-domain/DHPC complex are not very clear about the distance between Ca1 and the choline headgroup of PC. To clarify this issue, we have now added the distances between Ca1 and the DHPC choline nitrogen (5.5 Å) and between Ca1 and the DHPC phosphate (5.7 Å) to Table 1. Such distances are not indicative of interactions.

Is it possibly more favorable to have anionic lipid here (PS due to size or even PE due to size/shape)? The authors discuss this in Figure 4—figure supplement 1 by showing different lipid structures and discussing why they may or may not bind compared to PC.But, could the new structure here actually be revealing something more? Since PC has been studied as the bulk lipid that binds and PC was used to crystallize, perhaps the field has been biased against lipid mixtures (with small anionic lipid such as PS or other in the vesicles)?The authors suggest PS will have some repulsions due to the anionic headgroup but it is difficult to see this. Also, the phosphoryl group of anionic lipid could be stabilized by the calcium bound to the C2 domain.

To better illustrate the situation, we have revised Figure 2 by replacing panel C with a new space-filling rendition showing the C2-domain contacts that provide the especially ‘good fit’ for the phosphorylcholine headgroup of bound DHPC. The original panel C of Figure 2 has been incorporated into Figure 2—figure supplement 1. Moreover, we carried out additional new SPR experiments using wild-type C2-domain and vesicles composed of various phosphoglycerides including PS. The new data, which are now included in Figure 4, clearly show weak interaction by cPLA_2_α C2-domain for POPS, POPA, POPG, and POPI compared to POPC (subsection “Lipid specificity of cPLA_2_α C2-domian”). Unfortunately, technical problems prevented acquisition of reliable data for pure POPE as explained in the text. To circumvent the pure POPE problem, this lipid was mixed with POPS or POPC to assess the effect on C2-domain interaction. Whereas no interaction of PE could be detected in the presence of PC, a weak positive response was elicited in the presence of PS. Our findings are supported by earlier studies of refolded cPLA_2_α C-domain by Nalfeski et al., 1998, who reported strong binding to PC, but much weaker binding to vesicles composed of POPS and various other phosphoglycerides (POPG, POPA, mixed acyl PI, and POPE) using FRET and sedimentation assays.

Similarly, the PC headgroup is close to His62, which under some conditions could be partially protonated (~pKa near 6).

His62 appears to be sufficiently close for van der Waals contacts with the choline methyl groups. We initially suspected that His62 might be able to undergo weak cation-π interaction to the choline moiety (~ 5.0 Å). However, Ward et al., 2013, reported only slight differences in the binding of wild-type and H62A C2-domain to POPC vesicles using SPR. In this same study, changes in pH (6.0, 7.4, and 8.0) reportedly had no effect on binding to C2-domain to POPC vesicles. To clarify the situation, we have modified and added text along with the Ward et al., 2013, reference.

2) As written, the reduction in binding and/or activity by Y96A or N65D doesn't really prove they are involved in PC selectivity, it just demonstrated mutations reduce affinity and activity. To prove/demonstrate selectivity, one would have to compare the mutants and WT to lipids under these different conditions. The authors may want to consider rephrasing/rewriting these statements based upon their findings and that of the previous literature. For instance, although SM binding is low, what would the effects of mutations be? Would there be selective reduction in PC binding vs. that of SM?

We agree with the concern raised by the reviewer. We performed the suggested experiment involving testing of the mutants and WT with the sphingomyelin vesicles by SPR and have included the new results in Figure 4 (panel 4D). The data show that the relative changes in binding to POPC and 18:1-SM for WT and mutant C2-domains are similar (within experimental error) and modified the text accordingly (subsection “Lipid specificity of cPLA_2_α C2-domain”). We also have modified text by rephrasing to ‘phosphorylcholine headgroup lipids’ where needed.

Also, the SM vs. PC comparison is good and the authors are right that it hasn't been addressed much before. That being said, is SM relevant for Golgi or ER targeting of cPLA_2_ vs. that of PC or other lipids?

We have added new text providing information about the in vivo location of SM in the Golgi and the expected minimal impact on cPLA_2_α action (subsection “Lipid specificity of cPLA_2_α C2-domain” paragraph five).

3) Please address the following points regarding previous work in the field:The Introduction (paragraph three) points out that the lack of a lipid-bound crystal form of the C2 domain has hampered understanding of the structural basis of lipid activation mechanisms. While this current study does provide the lipid-bound structure to advance the field, it may be worth pointing out how the work of Ed Dennis and colleagues (among others) using molecular dynamics and hydrogen-deuterium exchange mass spectrometry has helped the field in understanding cPLA_2_ activity.

We agree with the recommendation and have added new text (Introduction paragraph four) as well as the relevant references (Burke et al., 2008; Cao et al., 2013; Mouchlisa et al., 2015) to improve the situation.

More details should be incorporated on the work done on Y96A and N65A by Bittova et al., 1999, as well as the role of N65A in membrane binding and penetration by Stahelin and Cho, 2001, and work by Burke et al. 2008, Bittova et al. studied the Y96A effects in full-length cPLA_2_ demonstrating a reduction in affinity for DHPC and a reduction in activity, which is confirmed in the current study. Similarly, it was shown that N65A reduced cPLA_2_ DHPC affinity and enzyme activity.

We agree with the recommendation and have added new text (Introduction paragraph two) as well as the relevant references (Bittova et al., 1999; Stahelin and Cho, 2001) to improve the situation. Also, new text is now modified and added to the beginning of the Discussion paragraph one.

4) Were the studies with full-length protein pursued with rat, human or chicken cPLA_2_? If not chicken, can the C2 chicken structure and lipid binding be directly compared to its influence on the C2-catalytic role in rat or human? If full-length chicken cPLA_2_ was used, how does its catalytic domain sequence compare to rat and human, and how does its activity compare to those published previously for the rat or human enzymes?

Clarifying text has been added to the Figure 3 legend. We also have included a new figure (Figure 1—figure supplement 1) showing the sequence alignment of the full-length cPLA_2_ of chicken, mouse, and human species. The very highly homologous nature of the sequences strongly suggests similar outcomes regardless of the species used.

Classically, the C2 domain of cPLA_2_ (rat) needed to be refolded from E. coli expression due to solubility issues. The authors have made great strides here in expressing the C2 domain from chicken cPLA_2_ in soluble form. This could be of interest to the field so a bit more detail on this solubility in the Materials and methods would be useful.

We thank the reviewer for acknowledging the technical advance that has accompanied our structural data. We have provided additional new text in the Materials and methods section. This new text describes the technical details associated with the development of C2-domain clones that can be expressed in soluble form in *E. coli*.

It would be helpful to have the alignment of the full cPLA_2_ of chicken with rat and human species.5) In Figure 5, where the authors model the crystal structure onto the schematic drawing and compare to EPR results, it would be more informative if distances are compared. For example, distances from x-ray structure, headgroup dimensions, EPR data can be compared. How deep in the membrane does the C2 domain go from experimental studies?

We agree and have now added the information regarding distances from the X-ray structure to the text describing the ad hoc model. We also discuss the C2-domain penetration depth within the context of liquid-crystalline bilayer structural information reported by Wiener and White, 1992. We find agreement with previously published penetration distances into the bilayer for Val97 and Ile39. The new text is now added in paragraph two of the Discussion section.

6) The binding of DHPC in one site is well appreciated but it is expected that the C2 domain will make contact and interact (at least non-specifically) with a number of lipid head groups around CBLs 1 and 3. Can this briefly be incorporated into the discussion model?

We have now incorporated this kind of descriptive text at the start of the Discussion section.

Also, in the crystal lattice, do the hydrophobic side chains of the lipid molecule interact with each other?

We have added information to the legend of Figure 1—figure supplement 2 (which shows the crystal packing lattice of the cPLA_2_ C2-domain/DHPC structural complex. The new information describes the distances associated between the acyl chains of the bound DHPC in the complexes. For comparison, we also have added the differing crystal packing lattice for lipid free C2-domain (PDB: 1RWL) as a new lower panel and referred to this new information in paragraph three of the Discussion section.

7) Synaptotagmin 1 C2A domain binds to three calcium ions. Can the authors comment and compare the structures?

Now included as Figure 6—figure supplement 1 is a direct structural comparison

(superpositioning) of the C2A domain of synaptotagmin-1 and the cPLA_2_α C-domain. The figure and associated legend, which provide a clear indication of differences and similarities with regards to Ca^2+^ binding, are pointed out in the Discussion subsection “Comparison of PL recognition with other C2–domains” paragraph three.

8) In the Materials and methods, the authors mention C1P purchased from Avanti but it is not clear when or how they were used in the paper? In crystallization attempts?

At the beginning of the Results, we briefly mentioned attempting to generate C2-domain complexes with various lipids including PC, related phosphoglycerides and C1P analogs.